# First Order Stochastic Optimization
# with Oblivious Noise

**Ilias Diakonikolas**
Department of Computer Sciences
University of Wisconsin-Madison
ilias@cs.wisc.edu

**Sushrut Karmalkar**
Department of Computer Sciences
University of Wisconsin-Madison
skarmalkar@wisc.edu

**Jongho Park**
KRAFTON
jongho.park@krafton.com

**Christos Tzamos**
Department of Informatics
University of Athens
tzamos@wisc.edu

## Abstract

We initiate the study of stochastic optimization with oblivious noise, broadly generalizing the standard heavy-tailed noise setup. In our setting, in addition to random observation noise, the stochastic gradient may be subject to independent *oblivious noise*, which may not have bounded moments and is not necessarily centered. Specifically, we assume access to a noisy oracle for the stochastic gradient of $f$ at $x$, which returns a vector $\nabla f(\gamma, x) + \xi$, where $\gamma$ is the bounded variance observation noise and $\xi$ is the oblivious noise that is independent of $\gamma$ and $x$. The only assumption we make on the oblivious noise $\xi$ is that $\mathbf{Pr}[\xi = 0] \geq \alpha$ for some $\alpha \in (0, 1)$. In this setting, it is not information-theoretically possible to recover a single solution close to the target when the fraction of inliers $\alpha$ is less than $1/2$. Our main result is an efficient *list-decodable* learner that recovers a small list of candidates, at least one of which is close to the true solution. On the other hand, if $\alpha = 1 - \epsilon$, where $0 < \epsilon < 1/2$ is sufficiently small constant, the algorithm recovers a single solution. Along the way, we develop a rejection-sampling-based algorithm to perform noisy location estimation, which may be of independent interest.

## 1  Introduction

A major challenge in modern machine learning systems is to perform inference in the presence of outliers. Such problems appear in various contexts, such as analysis of real datasets with natural outliers, e.g., in biology [Rosenberg et al., 2002, Paschou et al., 2010, Li et al., 2008], or neural network training, where heavy-tailed behavior arises from stochastic gradient descent when the batch size is small and when the step size is large [Hodgkinson and Mahoney, 2021, Gurbuzbalaban et al., 2021]. In particular, when optimizing various neural networks, there exists strong empirical evidence indicating that gradient noise frequently displays heavy-tailed behavior, often showing characteristics of unbounded variance. This phenomenon has been observed for fully connected and convolutional neural networks [Simsekli et al., 2019, Gurbuzbalaban and Hu, 2021] as well as attention models [Zhang et al., 2020]. Hence, it is imperative to develop robust optimization methods for machine learning in terms of both performance and security.

In this paper, we study robust first-order stochastic optimization under heavy-tailed noise, which may not have any bounded moments. Specifically, given access to a noisy gradient oracle for $f(x)$

37th Conference on Neural Information Processing Systems (NeurIPS 2023).

(which we describe later), the goal is to find a *stationary point* of the objective function $f : \mathbb{R}^d \to \mathbb{R}$, $f(x) := \mathbf{E}_\gamma[f(\gamma, x)]$, where $\gamma$ is drawn with respect to an arbitrary distribution on $\mathbb{R}^d$.

Previous work on $p$-heavy-tailed noise often considers the case where one has access to $\nabla f(\gamma, x)$ such that $\mathbf{E}_\gamma[\nabla f(\gamma, x)] = \nabla f(x)$ and $\mathbf{E}_\gamma[\|\nabla f(\gamma, x) - \nabla f(x)\|^p] \le \sigma^p$ for $p = 2$ [Nazin et al., 2019, Gorbunov et al., 2020], and recently for $p \in (1, 2]$ [Cutkosky and Mehta, 2021, Sadiev et al., 2023]. In contrast, Diakonikolas et al. [2019] investigates a more challenging noise model in which a small proportion of the sampled functions $f(\gamma, x)$ are arbitrary adversarial outliers. In their scenario, they have full access to the specific functions being sampled and can eventually eliminate all but the least harmful outlier functions. Their algorithm employs robust mean estimation techniques to estimate the gradient and filter out the outlier functions. Note that their adversarial noise model requires that at least half of the samples are inliers.

This work aims to relax the distributional assumption of heavy-tailed noise such that the fraction of outliers can approach one, and do not have any bounded moment constraints while keeping optimization still computationally tractable. Thus, we ask the following natural question:

*What is the weakest noise model for optimization in which efficient learning is possible, while allowing for strong corruption of almost all samples?*

In turn, we move beyond prior heavy-tailed noise models with bounded moments and define a new noise model for first-order optimization, inspired by the oblivious noise model studied in the context of regression [Bhatia et al., 2015, d'Orsi et al., 2021b]. We consider the setting in which the gradient oracle returns noisy gradients whose mean may not be $\nabla f(x)$, and in fact, may not even exist. The only condition on our additive noise distribution of $\xi$ is that with probability at least $\alpha$, oblivious noise takes the value zero. Without this, the optimization problem would become intractable. Notably, oblivious noise can capture any tail behavior that arises from independent $p$-heavy-tailed noise.

We now formally define the oblivious noise oracle below.

**Definition 1.1** (Oblivious Noise Oracle). *We say that an oracle $\mathcal{O}_{\alpha,\sigma,f}(x)$ is an oblivious-noise oracle, if, when queried on $x$, the oracle returns $\nabla f(\gamma, x) + \xi$ where $\gamma$ is drawn from some arbitrary distribution $Q$ and $\xi$ is drawn from $D_\xi$ independent of $x$ and $\gamma$, satisfying $\Pr_{\xi \sim D_\xi}[\xi = 0] \ge \alpha$. The distribution $Q$ satisfies $\mathbf{E}_{\gamma \sim Q}[\nabla f(\gamma, x)] = \nabla f(x)$ and $\mathbf{E}_{\gamma \sim Q}[\|\nabla f(\gamma, x) - \nabla f(x)\|^2] \le \sigma^2$.*

Unfortunately, providing an approximate stationary point $\widehat{x}$ such that $\|\nabla f(\widehat{x})\| \le \epsilon$, for some small $\epsilon > 0$, is information-theoretically impossible when $\alpha \le 1/2$. To see this, consider the case where $\Pr_\gamma[\gamma = 0] = 1$ and the function $f(0, x) = x^2$. If $\xi$ follows a uniform distribution over the set $\{-2, 0, 2\}$, when we query the oracle for a gradient at $x$, we will observe a uniform distribution over $\{2x - 2, 2x, 2x + 2\}$. This scenario cannot be distinguished from a similar situation where the objective is to minimize $f(0, x) = (x - 1)^2$, and $\xi$ follows a uniform distribution over $\{0, 2, 4\}$.

To address this challenge, we allow our algorithm to work in the *list-decodable* setting, implying that it may return a small list of candidate solutions such that one of its elements $\widehat{x}$ satisfies $\|\nabla f(\widehat{x})\| \le \epsilon$. We now define our main problem of interest.

**Definition 1.2** (List-Decodable Stochastic Optimization). *The problem of List-Decodable Stochastic Optimization with oblivious noise is defined as follows: For an $f$ that is $L$-smooth and has a global minimum, given access to $\mathcal{O}_{\alpha,\sigma,f}(\cdot)$ as defined in Definition 1.1, the goal is to output a list $\mathcal{L}$ of size $s$ such that $\min_{\widehat{x} \in \mathcal{L}} \|\nabla f(\widehat{x})\| \le \epsilon$.*

## 1.1 Our Contributions

As our main result, we demonstrate the equivalence between list-decodable stochastic optimization with oblivious noise and list-decodable mean estimation (LDME), which we define below.

**Definition 1.3** (List-Decodable Mean Estimation). *Algorithm $\mathcal{A}$ is an $(\alpha, \beta, s)$-LDME algorithm for $\mathcal{D}$ (a set of candidate inlier distributions) if with probability $1 - \delta_{\mathcal{A}}$, it returns a list $\mathcal{L}$ of size $s$ such that $\min_{\hat{\mu} \in \mathcal{L}} \|\hat{\mu} - \mathbf{E}_{x \sim D}[x]\| \le \beta$ for $D \in \mathcal{D}$ when given $m_{\mathcal{A}}$ samples $\{z_i + \xi_i\}_{i=1}^{m_{\mathcal{A}}}$ for $z_i \sim D$ and $\xi_i \sim D_\xi$ where $\Pr_{\xi \sim D_\xi}[\xi = 0] \ge \alpha$. If $1 - \alpha$ is a sufficiently small constant less than $1/2$, then $s = 1$.*

We define $\mathcal{D}_\sigma$ to be the following set of distributions over $\mathbb{R}^d$: $\mathcal{D}_\sigma := \{D \mid \mathbf{E}_D[\|x - \mathbf{E}_D[x]\|^2] \leq \sigma^2\}$. We also use $\tilde{O}(\cdot)$ to hide all log factors in $d, 1/\alpha, 1/\eta, 1/\delta$, where $\delta$ denotes the failure probability and $\eta$ is a multiplicative parameter we use in our algorithm.

Our first theorem shows that if there exists an efficient algorithm for the problem of list-decodable mean estimation and an inexact learner (an optimization method using inexact gradients), there is an efficient algorithm for list-decodable stochastic optimization.

**Theorem 1.4** (List-Decodable Stochastic Optimization → List-Decodable Mean Estimation). *Suppose that for any $f$ having a global minimum and being $L$-smooth, the algorithm $\mathcal{A}_G$, given access to $g_x$ satisfying $\|g_x - \nabla f(x)\| \leq O(\eta\sigma)$, recovers $\hat{x}$ satisfying $\|\nabla f(\hat{x})\| \leq O(\eta\sigma) + \epsilon$ in time $T_G$. Let $\mathcal{A}_{ME}$ be an $(\alpha, O(\eta\sigma), s)$-LDME algorithm for $\mathcal{D}_\sigma$. Then, there exists an algorithm (Algorithm 2) which uses $\mathcal{A}_{ME}$ and $\mathcal{A}_G$, makes $m = m_{\mathcal{A}_{ME}} + \tilde{O}(T_G \cdot (O(1)/\alpha)^{2/\eta}/\eta^5)$ queries to $\mathcal{O}_{\alpha,\sigma,f}$, runs in time $T_G \cdot \text{poly}(d, 1/\eta, (O(1)/\alpha)^{1/\eta}, \log(1/\delta\eta\alpha))$, and returns a list $\mathcal{L}$ of $s$ candidates satisfying $\min_{x \in \mathcal{L}} \|\nabla f(x)\| \leq O(\eta\sigma) + \epsilon$ with probability $1 - \delta_{\mathcal{A}_{ME}}$.*

To achieve this reduction, we develop an algorithm capable of determining the translation between two instances of a distribution by utilizing samples that are subject to additive noise. It is important to note that the additive noise can differ for each set of samples. As far as we know, this is the first guarantee of its kind for estimating the location of a distribution, considering the presence of noise, sample access, and the absence of density information.

**Theorem 1.5** (Noisy Location Estimation). *Let $\eta \in (0, 1)$ and let $D_\xi, D_z, D_{z'}$ be distributions such that $\Pr_{\xi \sim D_\xi}[\xi = 0] \geq \alpha$ and $D_z, D_{z'}$ are possibly distinct mean zero distributions with variance bounded by $\sigma^2$. Then there is an algorithm (Algorithm 5) which, for unknown $t \in \mathbb{R}^d$, takes $m = \tilde{O}((1/\eta^5)(O(1)/\alpha)^{2/\eta})$ samples $\{\xi_i + z_i + t\}_{i=1}^m$ and $\{\tilde{\xi}_i + z'_i\}_{i=1}^m$, where $\xi_i$ and $\tilde{\xi}_i$ are drawn independently from $D_\xi$ and $z_i$ and $z'_i$ are drawn from $D_z$ and $D_{z'}$ respectively, runs in time $\text{poly}(d, 1/\eta, (O(1)/\alpha)^{1/\eta}, \log(1/\delta\eta\alpha))$, and with probability $1 - \delta$ recovers $t'$ such that $|t - t'| \leq O(\eta\sigma)$.*

The fact that there exist algorithms for list-decodable mean estimation with the inliers coming from $\mathcal{D}_\sigma$ allows us to get concrete results for list-decodable stochastic optimization.

Conversely, we show that if we have an algorithm for list-decodable stochastic optimization, then we also get an algorithm for list-decodable mean-estimation. This in turn implies, via Fact 2.1, that an exponential dependence on $1/\eta$ is necessary in the list-size if we want to estimate the correct gradient up to an error of $O(\eta\sigma)$ when the inliers are drawn from some distribution in $\mathcal{D}_\sigma$.

**Theorem 1.6** (List-Decodable Mean Estimation → List-Decodable Stochastic Optimization). *Assume there is an algorithm for List-Decodable Stochastic Optimization with oblivious noise that runs in time $T$ and makes $m$ queries to $\mathcal{O}_{\alpha,\sigma,f}$, and returns a list $\mathcal{L}$ of size $s$ containing $\hat{x}$ satisfying $\|f(\hat{x})\| \leq \epsilon$. Then, there is an $(\alpha, \epsilon, s)$-LDME algorithm for $\mathcal{D}$ that runs in time $T$, queries $m$ samples, and returns a list of size $s$.*

If $\alpha = 1 - \epsilon$, where $0 < \epsilon < 1/2$ is at most a sufficiently small constant, the above theorems hold for the same problems, but with the constraint that the list is singleton.

## 1.2 Related Work

Given the extensive robust optimization and estimation literature, we focus on the most relevant work.

**Optimization with Heavy-tailed Noise** There is a wealth of literature on both the theoretical and empirical convergence behavior of stochastic gradient descent (SGD) for both convex and non-convex problems, under various assumptions on the stochastic gradient (see, e.g., Hardt et al. [2016], Wang et al. [2021] and references within). However, the noisy gradients have shown to be problematic when training ML models [Shen and Sanghavi, 2019, Zhang et al., 2020], hence necessitating robust optimization algorithms.

From a theoretical point of view, several noise models have been proposed to account for inexact gradients. A line of work [d'Aspremont, 2008, So, 2013, Devolder et al., 2014, Cohen et al., 2018] studies the effects of inexact gradients to optimization methods in terms of error accumulation. For instance, Devolder et al. [2014] demonstrates that, given an oracle that outputs an arbitrary

perturbation of the true gradient, the noise can be determined adversarially to encode non-smooth problems. Alternatively, many recent works [Lan, 2012, Gorbunov et al., 2020, Cutkosky and Mehta, 2021, Mai and Johansson, 2021, Sadiev et al., 2023] have studied $p$-heavy-tailed noise, an additive stochastic noise to the true gradient where one has access to $\nabla f(\gamma, x)$ such that $\mathbf{E}_\gamma[\nabla f(\gamma, x)] = \nabla f(x)$ and $\mathbf{E}_\gamma[\|\nabla f(\gamma, x) - \nabla f(x)\|^p] \leq \sigma^p$. For instance, Sadiev et al. [2023] propose and analyze a variant of clipped-SGD to provide convergence guarantees for when $p \in (1, 2]$. However, these noisy oracles, whether deterministic or stochastic, assume bounded norm or moments on the noise, an assumption that is not present in the oblivious noise oracle.

**Robust Estimation**   Our oblivious noise oracle for optimization is motivated by the recent work on regression under oblivious outliers. In the case of linear regression, the oblivious noise model can be seen as the weakest possible noise model that allows almost all points to be arbitrarily corrupted, while still allowing for recovery of the true function with vanishing error [Bhatia et al., 2015, Suggala et al., 2019]. This also captures heavy-tailed noise that may not have any moments. The setting has been studied for various problems, including online regression [Pesme and Flammarion, 2020], PCA [d'Orsi et al., 2021a], and sparse signal recovery [d'Orsi et al., 2022].

On the other hand, there has been a flurry of work on robust estimation in regards to worst-case adversarial outliers (see, e.g., Diakonikolas et al. [2016], Lai et al. [2016], Charikar et al. [2017], Diakonikolas et al. [2019]). Robust mean estimation aims to develop an efficient mean estimator when $(1 - \alpha)$-fraction of the samples is arbitrary. In contrast, list-decodable mean estimation generates a small list of candidates such that one of these candidates is a good estimator. While robust mean estimation becomes information-theoretically impossible when $\alpha \leq 1/2$, the relaxation to output a list allows the problem to become tractable for any $\alpha \in (0, 1]$ [Charikar et al., 2017, Diakonikolas et al., 2022]. See Diakonikolas and Kane [2019, 2023] for in-depth treatments of the subject.

In the past, robust mean estimators have been used to perform robust gradient estimation (see, e.g., Charikar et al. [2017], Diakonikolas et al. [2019]), which is similar to what we do in our paper. However, these results assume access to the entire function set, which allows them to discard outlier functions. In contrast, in our setting, we only have access to a noisy gradient oracle, so at each step, we get a fresh sample set and, hence, a different set of outliers. This introduces further difficulties , which we resolve via location estimation.

We note a subtle difference in the standard list-decodable mean estimation setting and the setting considered in this work. The standard list-decodable mean estimation setting draws an inlier with probability $\alpha$ and an outlier with the remaining probability. In contrast, our model gets samples of the kind $\xi + z$ where $z$ is drawn from the inlier distribution, and $\Pr[\xi = 0] > \alpha$, and is arbitrary otherwise. The algorithms for the mixture setting continue to work in our setting as well. Another difference is that the standard setting requires that the distribution have bounded variance *in every direction*. On the other hand, in the optimization literature, the assumption is that the stochastic gradient has bounded expected squared *norm* from the expectation.

**Location estimation**   Location estimation has been extensively studied since the 1960s. Traditional approaches to location estimation have focused on achieving optimal estimators in the asymptotic regime. The asymptotic theory of location estimation is discussed in detail in [Van der Vaart, 2000].

Recent research has attempted to develop the finite-sample theory of location estimation. These efforts aim to estimate the location of a Gaussian-smoothed distribution with a sample complexity that matches the optimal sample complexity up a constant (see [Gupta et al., 2022] and [Gupta et al., 2023]). However, these results assume prior knowledge of the likelihood function of the distribution up to translation.

Another closely related work initiates the study of robust location estimation for the case where the underlying high-dimensional distribution is symmetric and an $0 < \epsilon \ll 1/2$ fraction of the samples are adversarially corrupted Novikov et al. [2023]. This follows the line of work on robust mean estimation discussed above.

We present a finite-sample guarantee for the setting with noisy access to samples drawn from a distribution and its translation. Our assumption on the distribution is that it places an $\alpha$ mass at some point, where $\alpha \in (0, 1)$ and, for instance, could be as small as $1/d^c$ for some constant $c$. The noise is constrained to have mean zero and bounded variance, but crucially the noise added to the samples coming from the distribution and its translation might be drawn from *different distributions*. To the

best of our knowledge, this is the first result that has noisy access, does not have prior knowledge of the probability density of the distribution and achieves a finite-sample guarantee.

## 1.3 Technical Overview

For ease of exposition, we will make the simplifying assumption that the observation noise is bounded between $[-\sigma, \sigma]$. Let $y$ and $y'$ be two distinct mean-zero noise distributions, both bounded within the range $[-\sigma, \sigma]$. Define $\xi$ to be the oblivious noise drawn from $D_\xi$, satisfying $\Pr[\xi = 0] \geq \alpha$. Assume we have access to the distributions (i.e., we have infinite samples).

**Stochastic Optimization reduces to Mean Estimation.** In Theorem 1.4, we show how we can leverage a list-decodable mean estimator to address the challenge of list-decodable stochastic optimization with oblivious noise (see Algorithm 2). The key idea is to recognize that we can a generate list of gradient estimates, and update this list such that one of the elements always closely approximates the true gradient in $\ell_2$ norm at the desired point.

The algorithmic idea is as follows: First, run a list-decodable mean estimation algorithm on the noisy gradients at $x_0 = 0$ to retrieve a list $\mathcal{L}$ consisting of $s$ potential gradient candidates. This set of candidates contains at least one element which closely approximates $\nabla f(0)$.

One natural approach at this stage would be to perform a gradient descent step for each of the $s$ gradients, and run the list-decodable mean estimation algorithm again to explore all potential paths that arise from these gradients. However, this naive approach would accumulate an exponentially large number of candidate solutions. We use a location-estimation algorithm to tackle this issue.

We can express $f(\gamma, x)$ as $f(x) + e(\gamma, x)$ where $\mathbf{E}_\gamma[e(\gamma, x)] = 0$ and $\mathbf{E}_\gamma[\|e(\gamma, x)\|^2] < \sigma^2$. When we query $\mathcal{O}_{\alpha,\sigma,f}(\cdot)$ at $x$, we obtain samples of the form $\nabla f(x) + \xi + \nabla e(\gamma, x)$, i.e., samples from a translated copy of the oblivious noise distribution convolved with the distribution of $\nabla e(\gamma, x)$. We treat the distribution of $\nabla e(\gamma, x)$ as observation noise. The translation between the distributions of our queries to $\mathcal{O}_{\alpha,\sigma,f}(\cdot)$ at $x$ and $0$ correspond to $\nabla f(x) - \nabla f(0)$. To update the gradient, it is sufficient to recover this translation accurately. By doing so, we can adjust $\mathcal{L}$ by translating each element while maintaining the accuracy of the estimate.

Finally, we run a first-order learner for stochastic optimization using these gradient estimates. We select and explore $s$ distinct paths, with each path corresponding to an element of $\mathcal{L}$ obtained earlier. Since one of the paths always has approximately correct gradients, this path will converge to a stationary point in the time that it takes for the first-order learner to converge.

**Noisy Location Estimation in 1-D.** The objective of noisy location estimation is to retrieve the translation between two instances of a distribution by utilizing samples that are subject to additive mean-zero and bounded-variance noise.

In Lemma 3.1, we establish that if Algorithm 1 is provided with a parameter $\eta \in (0, 1)$ and samples from the distributions of $\xi + y$ and $\tilde{\xi} + y' + t$, it can recover the value of $t$ within an error of $O(\eta\sigma)$. Here $\xi$ and $\tilde{\xi}$ are independent draws from the same distribution.

Observe that in the absence of additive noise $y$ and $y'$, $t$ can be estimated exactly by simply taking the median of a sufficiently large number of pairwise differences between independent samples from $\xi$ and $\tilde{\xi} + t$. This is because the distribution of $\tilde{\xi} - \xi$ is symmetric at zero and $\Pr[\tilde{\xi} - \xi = 0] = \alpha^2$. However, with unknown $y$ and $y'$, this estimator is no longer consistent since the median of zero-mean and $\sigma$-variance random variables can be as large as $O(\sigma)$ in magnitude. In fact, Claim 3.2 demonstrates that for the setting we consider the median of $\xi + y$ can as far as $O(\sigma/\sqrt{\alpha})$ far from the mean of $\xi$. However, it does recover a rough estimate $t'_r$ such that $|t - t'_r| \leq O(\sigma/\sqrt{\alpha})$.

Consequently, our approach starts by using $t'_r$ as the estimate for $t$, and then iteratively improving the estimate by mitigating the heavy-tail influence of $\xi$. Note that if $\xi$ did not have heavy tails, then we could try to directly compute $t = \mathbf{E}[\xi + y' + t] - \mathbf{E}[\xi + y]$. Unfortunately, due to $\xi$'s completely unconstrained tail, $\xi$ may not even possess a well-defined mean. Nonetheless, if we are able to condition $\xi$ to a bounded interval, we can essentially perform the same calculation described above. In what follows, we describe one step of an iterative process to improve our estimate of $t$.

**Rejection Sampling.** Suppose we have an initial rough estimate of $t$, given by $t'_r$ such that $|t'_r - t| < A\sigma$. We can then re-center the distributions to get $\xi + z$ and $\xi + z'$, where $z$ and $z'$ have means of magnitude at most $A\sigma$, and have variance that is $O(\sigma)$. Claim 3.4 then shows that we can refine our estimate of $t$. It does so by identifying an interval $I = [-k\sigma, k\sigma]$ around $0$ such that the addition of either $z$ or $z'$ to $\xi$ does not significantly alter the distribution's mass within or outside of $I$. We show that since $z, z'$, and $\xi$ are independent, the $I$ that we choose satisfies $\mathbf{E}[\xi + z' \mid \xi + z' \in I] - \mathbf{E}[\xi + z \mid \xi + z \in I] = \mathbf{E}[\xi + z' \mid \xi \in I] - \mathbf{E}[\xi + z \mid \xi \in I] \pm \eta A = \mathbf{E}[z'] - \mathbf{E}[z] \pm \eta A$ for some $\eta < 1$.

To see that such an interval $I$ exists, we will show that for some $i$, there are pairs of intervals $(A\sigma i, (i+1)A\sigma]$ and $[-(i+1)A\sigma, -iA\sigma)$ which contain a negligible mass with respect to $\xi$. Since $z$ and $z'$ can move mass by at most $(A+1)\sigma$ with high probability, it is sufficient to condition on the interval around $0$ which is contained in this pair of intervals. Since we do not have access to $\xi$, we instead search for pairs of intervals of length $O(A\sigma)$ which have negligible mass with respect to both $\xi + z$ and $\xi + z'$, which suffices.

To do this, we will demonstrate an upper bound $\tilde{P}(i)$ on the mass crossing the $i^{th}$ pair of intervals described above. This will satisfy $\sum_i \tilde{P}(i) = C'$ for some constant $C'$.

To show that this implies there is an interval of negligible mass, we aim to demonstrate the existence of a $k \in \mathbb{N}$ such that $k\tilde{P}(k) \leq \eta$. We will do this through a contradiction. Suppose this is not the case for all $i \in [0, k]$, then $\sum_{i=0}^{k} \tilde{P}(i) \geq \eta \sum_{i=0}^{k}(1/i)$. If $k \geq \exp(10C'/\eta)$, we arrive at a contradiction because the right-hand side is at least $\eta \sum_{i=0}^{k}(1/i) \geq \eta \log(\exp(10C/\eta)) > 10C'$, while the left-hand side is bounded above by $C$.

Claim C.4 demonstrates, via a more involved and finer analysis, that for the bounded-variance setting where intervals very far away can contribute to mass crossing $A\sigma i$, there exists an $k$ such that $k\tilde{P}(k) \leq \eta \Pr[|\xi| < A\sigma k]$, and that taking the conditional mean restricted to the interval $[-kA\sigma, kA\sigma]$ allows us to improve our estimate of $t$. Here $\tilde{P}(k)$ denotes an upper bound on the total probability mass that crosses intervals described above.

**Extension to Higher Dimensions.** In order to extend the algorithm to higher dimensions, we apply the one-dimensional algorithm coordinate-wise, but in a randomly chosen coordinate-basis. Lemma 3.5 uses the fact that representing the distributions in a randomly rotated basis ensures that, with high probability, the inlier distribution will project down to each coordinate with a variance of $O(\sigma\sqrt{\log(d)}/\sqrt{d})$ to extend Lemma 3.1 to higher dimensions.

**Mean Estimation reduces to Stochastic Optimization.** In Theorem 1.6 we show that the problem of list-decodable mean estimation can be solved by using list-decodable stochastic optimization for the oblivious noise setting. This establishes the opposite direction of the reduction to show the equivalence of list-decodable mean estimation and list-decodable stochastic optimization.

The reduction uses samples from the list-decodable mean estimation problem to simulate responses from the oblivious oracle to queries at $x$. Let the mean of the inlier distribution be $\mu$. If the first-order stochastic learner queries $x$, we return $x + s$ where $s$ is a sample drawn from the list-decodable mean-estimation problem. These correspond to possible responses to the queries when $f(x) = (1/2)\|x + \mu\|^2$, where $\mu$ is the true mean. The first-order stochastic learner learns a $\widehat{\mu}$ from a list $\mathcal{L}$ such that $\|\nabla f(\widehat{\mu})\| = \|\widehat{\mu} + \mu\| \leq \epsilon$; then the final guarantee of the list-decodable mean estimator follows by returning $-\mathcal{L}$ which contains $-\widehat{\mu}$.

## 2 Preliminaries

**Basic Notation** For a random variable $X$, we use $\mathbf{E}[X]$ for its expectation and $\mathbf{Pr}[X \in E]$ for the probability of the random variable belonging to the set $E$. We use $\mathcal{N}(\mu, \sigma^2)$ to denote the Gaussian distribution with mean $\mu$ and variance matrix $\sigma^2$. When $D$ is a distribution, we use $X \sim D$ to denote that the random variable $X$ is distributed according to $D$. When $S$ is a set, we let $\mathbf{E}_{X \sim S}[\cdot]$ denote the expectation under the uniform distribution over $S$. When clear from context, we denote the empirical expectation and probability by $\widehat{\mathbf{E}}$ and $\widehat{\mathbf{Pr}}$. We denote $\|\cdot\|$ as the $\ell_2$-norm and assume $f : \mathbb{R}^d \to \mathbb{R}$ is differentiable and $L$-smooth, i.e., $\|\nabla f(x) - \nabla f(x')\| \leq L\|x - x'\|$ for all $x, x' \in \mathbb{R}^d$.

$\xi$ will always denote the oblivious noise drawn from a distribution $Q$ satisfying $\mathbf{Pr}_{\xi \sim Q}[\xi = 0] \geq \alpha$. $y, y'$ will be used to denote mean-zero and variance at-most $\sigma^2$ random variables. Also define $e(\gamma, x) := f(\gamma, x) - f(x)$ and the interval $\sigma(a, b] := (\sigma a, \sigma b]$.

**Facts** We use these algorithmic facts in the following sections. We use a list-decodable robust mean estimation subroutine in a black-box manner. The proof for list-decodable mean estimation can be found in Appendix D, while such algorithms can be found in prior work, see, e.g., Charikar et al. [2017], Diakonikolas et al. [2020a]. We also define a $(\beta, \epsilon)$-inexact-learner for $f$.

**Fact 2.1** (List-decoding algorithm). *There is an $(\alpha, \sigma\eta, \tilde{O}((1/\alpha)^{1/\eta^2}))$-LDME algorithm for the an inlier distribution belonging to $\mathcal{D}_\sigma$ which runs in time $\tilde{O}(d(1/\alpha)^{1/\eta^2})$ and succeeds with probability $1 - \delta$. Conversely, any algorithm which returns a list, one of which makes an error of at most $O(\eta\sigma)$ in $\ell_2$ norm to the true mean, must have a list whose size grows exponentially in $1/\eta$.*

*If $1 - \alpha$ is a sufficiently small constant less than half, then the list size is 1 to get an error of $O(\sqrt{1 - \alpha}\, \sigma)$.*

We now define the notion of a robust-learner, which is an algorithm which, given access to approximate gradients is able to recover a point at which the gradient norm is small.

**Definition 2.2** (Robust Inexact Learner). *Let $f$ be a $L_s$-smooth function with a global minimum and $\mathcal{O}_{\beta,f}^{\text{grad}}(\cdot)$ be an oracle which when queried on $x$ returns a vector $g_x$ satisfying $\|g_x - \nabla f(x)\| \leq \beta$. We say that an algorithm $\mathcal{A}_G$ is an $(\beta, \epsilon)$-inexact-learner for $f$ if, given access to $\mathcal{O}_{\beta,f}^{\text{grad}}$, the algorithm $\mathcal{A}_G$ recovers $\hat{x}$ satisfying $\|\nabla f(\hat{x})\| \leq \beta + \epsilon$. We will assume $\mathcal{A}_G$ runs in time $T_G$.*

Several algorithms for convex optimization amount to there being an existence of a robust learner in the convex setting. More recently, for smooth nonconvex optimization, the convergence result for SGD under inexact gradients due to Ajalloeian and Stich [2020] doubles as a $(\beta, \epsilon)$-inexact-learner running in $T_G = O(LF/(\beta + \epsilon)^2)$ iterations, where $F = f(x_0) - \min_x f(x)$ and $x_0$ is the initial point. This follows by an application of Theorem 4 from Ajalloeian and Stich [2020] and by taking the point in the set of iterates that minimizes the gradient norm.

## 3 Location Estimation

To estimate how the gradient changes between iterations, we will need to estimate the shift between a distribution and its translation. In this section, we give an algorithm which, given access to samples from a distribution and its noisy translation returns an estimate of $t$ accurate up to an error of $O(\eta\sigma)$ in $\ell_2$-norm. We show a proof sketch here, and a more detailed proof of all the claims involved in Appendix C.

**Lemma 3.1** (One-dimensional location-estimation). *There is an algorithm (Algorithm 1) which, for $m = (1/\eta^5)(O(1)/\alpha)^{2/\eta} \log(1/\eta\alpha\delta)$, given samples $\{\xi_i + y_i + t\}_{i=1}^m$ and $\{\tilde{\xi}_i + y'_i\}_{i=1}^m$ where $\xi_i$ and $\tilde{\xi}_i$ are both drawn from $D_\xi$, $y_i$ and $y'_i$ are drawn from distinct distributions with mean-zero with variance bounded above by $\sigma$ and $t \in \mathbb{R}$ is an unknown translation, runs in time $\tilde{\text{poly}}(1/\eta, (O(1)/\alpha)^{1/\eta}, \log(1/\delta\eta\alpha))$ and recovers $t'$ such that $|t - t'| \leq O(\eta\sigma)$.*

*Proof.* We first identify $t$ up to an error of $O(\sigma/\sqrt{\alpha})$ with the following claim.

**Claim 3.2** (Rough Estimate). *There is an algorithm which, for $m = O((1/\alpha^4)\log(1/\delta))$ given samples $\{\xi_i + y_i + t\}_{i=1}^m$ and $\{\tilde{\xi}_i + y'_i\}_{i=1}^m$ where $\xi_i$ and $\tilde{\xi}_i$ are both drawn from $D_\xi$, $y_i$ and $y'_i$ are drawn from distinct distributions with mean-zero with variance bounded above by $\sigma$ and $t \in \mathbb{R}$ is an unknown translation, returns $|t'_r - t| \leq O(\sigma\alpha^{-1/2})$.*

We use $t'_r$ to center the two distributions up to an error of $\sigma\alpha^{-1/2}$. Let the centered distributions be given by $\xi + z$ and $\xi + z'$ where $z, z'$ are independent, have means that are at most $\sigma\alpha^{-1/2}$ in magnitude and have variance that is bounded by $4\sigma^2$.

Let $A\Delta B$ denote the symmetric difference of the two events $A, B$. To improve our estimate further, we restrict our attention to an interval $I = \sigma(-i, i)$ such that $\mathbf{Pr}[(\xi + z \in I)\,\Delta\,(\xi \in I)] <$

---

**Algorithm 1** One-dimensional Location Estimation: $\text{Shift1D}(S_1, S_2, \eta, \sigma, \alpha)$

---

**Input:** Sample sets $S_1, S_2 \subset \mathbb{R}^d$ of size $m$, $\alpha, \eta \in (0,1)$, $\sigma > 0$

1. Let $T = O(\log_{1/\eta}(1/\alpha))$. For $j \in \{1, 2\}$, partition $S_j$ into $T$ equal pieces, $S_j^{(i)}$ for $i \in [T]$.
2. $D = \{a - b \mid a \in S_1^{(1)}, b \in S_2^{(1)}\}$.
3. $t'(1) := \text{median}(D)$.
4. Set $A = O(1/\sqrt{\alpha})$.
5. Repeat steps 6 to 12, for $i$ going from 2 to $T$:
6. $S_1^{(i)} := S_1^{(i)} - t'_r(i-1)$.
7. For $j \in \{1, 2\}$

$$\hat{P}_j(i) := O(1) \Pr_{x \sim S_j^{(i)}}[|x| \in A\sigma(i-5, i+5)]$$

$$+ O(1) \sum_{j=1}^{i-1} (1/(i-j)^2) \Pr_{x \sim S_j^{(i)}}[|x| \in Aj\sigma + A\sigma[-4, 5).]$$

8. Let $\hat{P}(i) = \hat{P}_1(i) + \hat{P}_2(i)$.
9. Identify an integer $k \in [(1/\alpha\eta), (C/\alpha + 1/(\alpha\eta)^\eta)^{1/\eta}]$ such that

$$\hat{P}(k) \leq \eta \sum_{j \in \{1,2\}} \Pr_{x \sim S_j^{(i)}}[|x| \in A\sigma k] \pm O(\eta/i).$$

10. $t'(i) := t'(i-1) + \mathbf{E}_{z \sim S_1^{(i)}}[z \mid |z| \leq A\sigma k] - \mathbf{E}_{z \sim S_2^{(i)}}[z \mid |z| \leq A\sigma k]$.
11. $A := \eta A$.
12. Return $t'(T)$

---

$O(\eta \Pr[\xi \in I])$ and $\Pr[(\xi + z' \in I) \, \Delta \, (\xi \in I)] < O(\eta \Pr[\xi \in I])$, i.e., neither $z$ nor $z'$ moves more than a negligible mass of $\xi$ either into, or out of $I$.

To detect such an interval (if it exists), we control the total mass contained in, and moved across the intervals $\sigma(-i-1, -i)$ and $\sigma(i, i+1)$ when $z$ or $z'$ are added to $\xi$. We denote this mass by $P''(i)$. Claim C.4 shows that there is a function $\tilde{P}(\cdot)$ which we can compute using our samples from the distributions of $\xi + z$ and $\xi + z'$, which serves as a good upper bound on $P''(i)$.

**Claim 3.3.** *There exists a function $\tilde{P} : \mathbb{N} \to \mathbb{R}^+$ which satisfies the following:*
*1. For all $i \in \mathbb{N}$, $\tilde{P}(i) \geq P(i, z) + P(i, z')$ which can be computed using samples from $\xi + z$ and $\xi + z'$.*
*2. There is a $k \in [(1/\alpha\eta), (C/\alpha + 1/(\alpha\eta)^\eta)^{1/\eta}]$ such that $k\tilde{P}(k) < \eta \sum_{j=0}^{k} \tilde{P}(k)$.*
*3. $\sum_{j=0}^{k} \tilde{P}(k) = O(\Pr[|\xi + z| \leq A\sigma k] + \Pr[|\xi + z'| \leq A\sigma k])$.*
*4. With probability $1 - \delta$, for all $i < (O(1)/\alpha)^{1/\eta}/\eta$, $\tilde{P}(i)$ can be estimated to an accuracy of $O(\eta/i)$ by using $(O(1)/\alpha)^{2/\eta} \log(1/\eta\alpha\delta)/\eta^5$ samples from the distributions of $\xi + z$ and $\xi + z'$.*

Claim 3.4 shows that if such a $\tilde{P}(\cdot)$ exists, then the difference of the conditional expectations suffices to get a good estimate. This is because conditioning on $|\xi + z| < 10k\sigma$ satisfying conditions (2) and (3) above is almost the same as conditioning on $|\xi| < 10k\sigma$.

**Claim 3.4.** *Suppose $i > (\eta\alpha)^{-1}$ and $\tilde{P}(\cdot)$ satisfies the conclusions of Claim C.4. If $z, z'$ have $A\sigma$ bounded means and $4\sigma^2$ bounded variances for some A, then $\mathbf{E}[\xi + z \mid |\xi + z| \leq A\sigma i] - \mathbf{E}[\xi + z' \mid |\xi + z'| \leq A\sigma i] = \mathbf{E}[z] - \mathbf{E}[z'] \pm O(A\eta\sigma)$*

The conclusion of the one-dimensional location estimation theorem then follows by putting these together, and iterating the above steps after re-centering the means. The sample complexity is governed by the following considerations:

(1) We require the $\mathbf{E}[\xi + z \mid |\xi + z| \le \sigma\alpha^{-1/2}k] - \mathbf{E}[\xi + z' \mid |\xi + z'| \le \sigma\alpha^{-1/2}k]$ to concentrate around its mean for some $i \le \sigma(C/\alpha + 1/(\alpha\eta)^\eta)^{1/\eta}$. (2) We require $\mathbf{Pr}[\xi + z \in \sigma(i, i+1)]$ to concentrate for all integer $i$ of magnitude at most $\sigma(C/\alpha + 1/(\alpha\eta)^\eta)^{1/\eta}$.

An application of Hoeffding's inequality (Lemma A.1) and a union bound over all the events prove that it suffices to ensure sample complexity to be $m \ge \text{poly}(1/\eta, (O(1)/\alpha)^{1/\eta}, \log(1/\delta\eta\alpha))$.

$\square$

The following Lemma 3.5 uses the algorithm for one dimension to derive a higher-dimensional guarantee. The idea is to perform a random rotation, which, with high probability, ensures that the variance of the distribution is $O(\sigma\sqrt{\log(d)}/\sqrt{d})$ along each basis and then apply the one-dimensional algorithm coordinate-wise according to the random bases. We defer the proof to Appendix C. The algorithm referenced in this lemma is named "ShiftHighD", which we use later.

**Lemma 3.5** (Location Estimation). *Let $D_{y_i}$ for $i \in \{1, 2\}$ be the distributions of $\xi + z_i$ for $i \in \{1, 2\}$ where $\xi$ is drawn from $D_\xi$ and $\mathbf{Pr}_{\xi \sim D_\xi}[\xi = 0] \ge \alpha$ and $z_i \sim D_i$ are distributions over $\mathbb{R}^d$ satisfying $\mathbf{E}_{D_i}[x] = 0$ and $\mathbf{E}_{D_i}[\|x\|^2] \le \sigma^2$. Let $v \in \mathbb{R}^d$ be an unknown shift, and $D_{y_2,v}$ denote the distribution of $y_2$ shifted by $v$. There is an algorithm (Algorithm 5), which draws $\text{poly}(1/\eta, (O(1)/\alpha)^{1/\eta}, \log(1/\delta\eta\alpha))$ samples each from $D_{y_1}$ and $D_{y_2,v}$ runs in time $\text{poly}(d, 1/\eta, (O(1)/\alpha)^{1/\eta}, \log(1/\delta\eta\alpha))$ and returns $v'$ satisfying $\|v' - v\| \le O(\eta\sigma)$ with probability $1 - \delta$.*

# 4 Algorithmic Results for Stochastic Optimization

Our first main result demonstrates that if we have access to an algorithm that recovers a list of size $s$ such that one of the means is $O(\eta\sigma)$-close to the true mean in $\ell_2$ norm, then there is an algorithm which is able to recover $\hat{x}$ such that $\|\nabla f(\hat{x})\| \le O(\eta\sigma) + \epsilon$ given access to gradients that are close to the true gradient up to an $\ell_2$ error of $O(\eta\sigma)$.

## 4.1 Proof of Theorem 1.4: List-decodable Stochastic Optimization → LDME

---

**Algorithm 2** Noisy Gradient Optimization: NoisyGradDesc$(\alpha, \tau, \delta, \mathcal{O}, \mathcal{A}_G, \mathcal{A}_{ME})$

---

**Input:** $\alpha, \eta, \delta, (\alpha, O(\eta\sigma), s)$-LDME algorithm $\mathcal{A}_{ME}$ for $\mathcal{D}_\sigma, \mathcal{O}_{\alpha,\sigma,f}(\cdot), (O(\eta\sigma), \epsilon)$-learner $\mathcal{A}_G$.

1. Let $m' = m_{\mathcal{A}_{ME}}$
2. Query $\mathcal{O}_{\alpha,\sigma,f}(0)$ to get samples $S^* \leftarrow \{\nabla f(\gamma_i, 0) + \xi_i \mid i \in [m']\}$.
3. Let $\mathcal{L}_0 := \{g_1, \dots, g_s\} \leftarrow \mathcal{A}_{ME}(S)$.
4. Initialize starting points $x_i^0 := g_i$ for $i \in [s]$.
5. For each $i \in [s]$, run $\mathcal{A}_G$ with $x_i^0$ as the initial point and InexactOracle$(x; \mathcal{O}_{\alpha,\sigma,f}(\cdot), O(\eta\sigma), \mathcal{L}_0)_i$ (the $i$-th element of the list) as gradient access to output $x_i^{final}$.
6. Return $\{x_i^{final} \mid i \in [s]\}$.

---

*Proof.* The key to our proof is to recognize that at every step, we effectively have an oracle for an inexact gradient of $f$ in the sense of $\mathcal{O}_{O(\eta\sigma),f}^{\text{grad}}$ as defined in Definition 2.2.

---

**Algorithm 3** Inexact Gradient Oracle: InexactOracle$(x; \mathcal{O}_{\alpha,\sigma,f}, \tau, \mathcal{L}_0)$

---

**input:** $x$, oracle $\mathcal{O}_{\alpha,\sigma,f}$, error $\tau$, a list $\mathcal{L}_0$ of candidates such that $\min_{g \in \mathcal{L}} \|g - \nabla f(0)\| \le O(\eta\sigma)$.
1. Let $\eta := \sigma/\tau$ and $m' = \tilde{O}((1/\eta\alpha)^{3/\eta})$
2. Query $\mathcal{O}_{\alpha,\sigma,f}(0)$ to get samples $S^* \leftarrow \{\nabla f(\gamma_i, 0) + \xi_i \mid i \in [m']\}$.
3. Query $\mathcal{O}_{\alpha,\sigma,f}(x)$ to get samples $S \leftarrow \{\nabla f(\gamma_i, x) + \tilde{\xi}_i \mid i \in [m']\}$.
4. $v := \text{ShiftHighD}(S, S^*, \eta, \sigma)$.
5. Return $\mathcal{L}_0 + v$.

---

**Claim 4.1.** *Given an initial list $\mathcal{L}_0 := \{g_1, \ldots, g_s\}$ such that there is some fixed $i \in [s]$ for which $\|g_i - \nabla f(0)\| \leq O(\eta\sigma)$, a point $x$, access to $\mathcal{O}_{\alpha,\sigma,f}(\cdot)$, Algorithm 3 returns a list $\mathcal{L} := \{g_1', \ldots, g_s'\}$ such that $\|g_i' - \nabla f(x)\| \leq O(\eta\sigma)$ for the same $i$.*

*Proof.* $\mathcal{O}_{\alpha,\sigma,f}(y)$ returns samples of the kind $\{\xi_i + e(\gamma,x)_i + \nabla f(x)\}_{i=1}^m$ and $\{\tilde{\xi}_i + e(\gamma,0)_i + \nabla f(0)\}_{j=1}^m$ for $y = x$ and $y = 0$, with $e(\gamma,y)_i$ being drawn from a distribution with $0$ and variance bounded by $\sigma^2$ and $\xi_i$ and $\tilde{\xi}_i$ being drawn from $D_\xi$ where $\mathbf{Pr}_{\xi \sim D_\xi}[\xi = 0] \geq \alpha$. Hence, one can interpret the samples drawn as being in the setting of Lemma 3.5 with the shift $v = \nabla f(0) - \nabla f(x)$.

Let $S_1$ and $S_2$ be drawn from $\mathcal{O}_{\alpha,\sigma,f}(y)$ with $y = 0$ and $y = x$. Running Algorithm 5 on $S_1, S_2$, we recover $v$ satisfying $\|v - (\nabla f(x) - \nabla f(0))\| \leq O(\eta\sigma)$. A triangle inequality now tells us that if we set $g_i' := g_i + v$, we get $\|g_i' - \nabla f(x)\| = \|g_i' - g_i + g_i - \nabla f(0) + \nabla f(0) - \nabla f(x)\| \leq O(\eta\sigma) + O(\eta\sigma) = O(\eta\sigma)$. □

$g_i'$ can be interpreted as the output of $O^{\text{grad}}_{O(\eta\sigma),f}(x)$, since $\|g_i' - \nabla f(x)\| \leq O(\eta\sigma)$. The result then follows from the guarantees of $\mathcal{A}_G$, since for at least one of the sequences of gradients, Algorithm 2 replicates every step of $\mathcal{A}_G$. □

Substituting the guarentees of Fact 2.1 for the list-decoding algorithm in the above theorem then gives us the following corollary, the proof of which we defer to Appendix E.

**Corollary 4.2.** *Given access to oblivious noise oracle $\mathcal{O}_{\alpha,\sigma,f}$, a $(O(\eta\sigma),\epsilon)$-inexact-learner $\mathcal{A}_G$ running in time $T_G$, there exists an algorithm which takes $\text{poly}((O(1)/\alpha)^{1/\eta^2}, \log(T_G/\delta\eta\alpha))$ samples, runs in time $T_G \cdot \text{poly}(d, (O(1)/\alpha)^{1/\eta^2}, \log(1/\eta\alpha\delta))$, and with probability $1 - \delta$ returns a list $\mathcal{L}$ of size $\tilde{O}((1/\alpha)^{1/\eta^2})$ such that $\min_{x \in \mathcal{L}} \|\nabla f(x)\| \leq O(\eta\sigma) + \epsilon$. Additionally, the exponential dependence on $1/\eta$ in the size of the list is necessary.*

### 4.2 Proof of Theorem 1.6: LDME → List-Decodable Stochastic Optimization

In this subsection, we show the converse of the results from the previous subsection, i.e. that list-decodable stochastic optimization can be used to perform list-decodable mean estimation.

*Proof.* Assume there exists an algorithm $\mathcal{A}$ that can recover a $s$-sized list containing $\widehat{x}$ such that $\|\nabla f(\widehat{x})\| \leq \epsilon$ when given access to $\mathcal{O}^o_{\alpha,\sigma,f}(\cdot)$. From the list-decodable mean estimation setting, denote $D$ to be the distribution of $\xi + z$ where $\mathbf{E}[z] = \mu$ and $\mathbf{E}[\|z - \mu\|^2] < \sigma^2$, and $\mathbf{Pr}[\xi = 0] \geq \alpha$.

The goal of LDME is to recover $\mu$ from samples from $D$. We will show that we can recover an $s$-sized list that contains a $\widehat{\mu}$ satisfying $\|\widehat{\mu} - \mu\| \leq \epsilon$.

We do this by simulating the oracle $\mathcal{O}_{\alpha,\sigma,(1/2)\|x+\mu\|^2}(\cdot)$ for $\mathcal{A}$. To do this, whenever $\mathcal{A}$ asks for a query at $x$, we return $x + p$ where we sample $p \sim D$ of the list-decodable setting. This effectively simulates the oracle for the objective function $f(x) = (1/2)\|x + \mu\|^2$, where the oblivious noise $\xi$ is the same, and the observation noise is $(z - \mu)$. Hence $\mathcal{A}$ will return a list $\mathcal{L}$ of size $s$ containing $\widehat{x}$ satisfying $\|\widehat{x} + \mu\| \leq \epsilon$. Finally, return $-\mathcal{L}$. This is the solution to list-decodable mean estimation because $-\mathcal{L}$ contains $\widehat{\mu} := -\widehat{x}$, which satisfies $\|\widehat{\mu} - \mu\| \leq \epsilon$. □

## 5 Conclusion

In this paper, we have initiated the study of stochastic optimization in the presence of oblivious noise, which extends the traditional heavy-tailed noise framework. In our setting, the stochastic gradient is additionally affected by independent oblivious noise that lacks bounded moments and may not be centered. We have also designed an algorithm for finite-sample noisy location estimation based on taking conditional expectations, which we believe is of independent interest. We note that while the exponential dependence on $1/\eta$ for the size of the list is unavoidable, but it is an open problem to show that this is the case for the problem of noisy location estimation.

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

## Supplementary Material

**Organization** In Appendix A, we state some elementary probabilistic facts. The next two sections focus on proving our lemma on noisy location estimation. In Appendix B, we prove some critical lemmas used in the proof, and in Appendix C, we present the complete version of our location estimation algorithm, while addressing some typos in the main text. We mention some typos in footnotes and correct other minor typos without a mention.

Moving forward, in Appendix D, we introduce an algorithm and prove a hardness result for the specific version of list-decodable mean estimation we consider, which differs from prior work. Finally, in Appendix E, we state the final guarantees we can get for the problem of list-decodable stochastic optimization, incorporating our lemma from Appendix D.

## A    Elementary Probability Facts

In this section, we recall some elementary lemmas from probability theory.

**Lemma A.1** (Hoeffding). *Let $X_1, \ldots X_n$ be independent random variables such that $X_i \in [a_i, b_i]$. Let $S_n := \frac{1}{n} \sum_{i=1}^n X_i$, then for all $t > 0$*

$$\mathbf{Pr}[|S_n - \mathbf{E}[S_n]| \geq t] \leq \exp\left(-\frac{2n^2 t^2}{\sum_{i=1}^n (b_i - a_i)^2}\right) .$$

**Lemma A.2** (Multivariate Chebyshev). *Let $X_1, \ldots, X_m$ be independent random variables drawn from $D$ where $D$ is a distribution over $\mathbb{R}^d$ such that $\mathbf{E}_{X \sim D}[X] = 0$ and $\mathbf{E}_{X \sim D}[\|X\|^2] \leq \sigma^2$. Let $S_m := \frac{1}{m} \sum_{i=1}^m X_i$, then for all $t > 0$*

$$\mathbf{Pr}[\|S_m\| \geq t] \leq \sigma^2 / m t^2 .$$

*Proof.* We first prove the following upper bound,

$$\mathbf{Pr}\left[\forall v \, \|v\| = 1. \left|\frac{1}{m} \sum_{i \in [m]} X_i \cdot v\right| > t\right] < \frac{\mathbf{E}[|\sum_{i \in [m]} X_i \cdot v|^2]}{m^2 \, t^2} < \frac{\mathbf{E}[\sum_{i,j} (X_i \cdot v)(X_j \cdot v)]}{m^2 \, t^2}$$

$$< \frac{\sum_{i,j} \mathbf{E}[(X_i \cdot v)(X_j \cdot v)]}{m^2 \, t^2} < \frac{\sum_i \mathbf{E}[(X_i \cdot v)^2]}{m^2 \, t^2} < \frac{\sum_i \mathbf{E}[\|X_i\|^2]}{m^2 \, t^2}$$

$$< \frac{m\sigma^2}{m^2 \, t^2} = \frac{\sigma^2}{m \, t^2} .$$

Since the inequality holds for all unit $v$, it also holds for the unit $v$ in the direction of $S_m$, completing the proof. □

**Fact A.3** (Inflation via conditional probability). *Let $y$ be a random variable with mean $\mu$ and variance $\sigma^2$ and let $\xi$ be an arbitrary random variable independent of $y$, then*

$$\mathbf{Pr}[\xi \in (a, b)] < (1 + 2/A^2) \, \mathbf{Pr}[\xi + y \in (a + \mu - \sigma A, b + \mu + \sigma A)$$
$$< (1 + 2/A^2) \, \mathbf{Pr}[\xi + y \in (a - |\mu| - |\sigma A|, b + |\mu| + |\sigma A|)] .$$

*Proof.* To do this, we inflate the intervals and use conditional probabilities.

$$\mathbf{Pr}[\xi \in (a, b)] = \mathbf{Pr}[\xi + y \in (a + \mu - \sigma A, b + \mu + \sigma A) \mid |y - \mu| < \sigma A]$$
$$= \frac{\mathbf{Pr}[\xi + y \in (a + \mu - \sigma A, b + \mu + \sigma A) \text{ and } |y - \mu| < \sigma A]}{\mathbf{Pr}[|y - \mu| < \sigma A]}$$
$$< (1 - 1/A^2)^{-1} \, \mathbf{Pr}[\xi + y \in (a + \mu - \sigma A, b + \mu + \sigma A) \text{ and } |y - \mu| < \sigma A]$$
$$< (1 + 2/A^2) \, \mathbf{Pr}[\xi + y \in (a + \mu - \sigma A, b + \mu + \sigma A)] .$$

The second inequality above follows from observing that we are simply lengthening the interval.

We will often use the second version for ease of analysis. □

# B  Useful Lemmas

In this section, we present some helpful lemmas for the algorithm on noisy one-dimensional location estimation.

To recap the setting: We can access samples from distributions $\xi + y$ and $\xi + y' + t$. Here, $\mathbf{Pr}[\xi = 0] > \alpha$, $y$ and $y'$ are distributions with zero mean and bounded variance, and $t \in \mathbb{R}$ is an unknown translation. Our objective is to estimate the value of $t$.

## B.1  Useful Lemma for Rough Estimation

Our algorithm for one-dimensional location estimation consists of two steps. In the first step, we obtain an initial estimate of the shift between the two distributions by computing pairwise differences of samples drawn from each distribution. This involves taking the median of the distribution of $x + y$, where $x$ is symmetric and $y$ has mean 0 and bounded variance.

The following lemma demonstrates that the median of this distribution is at most $O(\sigma\alpha^{-1/2})$, where $\sigma$ is the standard deviation of $y$. Furthermore, this guarantee cannot be improved.

**Fact B.1** (Median of Symmetric + Bounded-variance Distribution). *Let $x$ be a random variable symmetric around 0 such that $\alpha \in (0, 1)$, $\mathbf{Pr}[x = 0] \geq \alpha$. Let $y$ be a random variable with mean 0 and variance $\sigma^2$. If $S$ is a set of $O(1/\alpha^2 \log(1/\delta))$ samples drawn from the distribution of $x + y$, $|\mathrm{median}(S)| \leq O(\sigma\alpha^{-1/2})$.*

*This guarantee is tight in the sense that there exist distributions for $x$ and $y$ satisfying the above constraints, such that* $\mathrm{median}(x + y)$ *can be as large as* $\Omega(\sigma\alpha^{-1/2})$.

*Proof.* We show that $\mathbf{Pr}[x + y < -O(\sigma/\sqrt{\alpha})] < 0.5$ and $\mathbf{Pr}[x + y > O(\sigma/\sqrt{\alpha})] < 0.5$, as a result, $|\mathrm{median}(x + y)| < O(\sigma/\sqrt{\alpha})$. We will later transfer this guarantee to the uniform distribution over the samples.

Applying Fact A.3 to the first probability, we see that $\mathbf{Pr}[x + y < -O(\sigma/\sqrt{\alpha})] < (1 + \alpha)\mathbf{Pr}[x < -O(\sigma/\sqrt{\alpha})]$.

Since $\mathbf{Pr}[x = 0] \geq \alpha$, we see that $\mathbf{Pr}[x < 0] \leq 1/2 - \alpha$,

and so $\mathbf{Pr}[x + y < -O(\sigma/\sqrt{\alpha})] < (1 + \alpha)(0.5 - \alpha) = 0.5 - \alpha + 0.5\alpha - \alpha^2 = 0.5 - 0.5\alpha - \alpha^2 < 0.5$.

The upper bound follows similarly.

Since $\mathbf{Pr}[x = 0] \geq \alpha$ and $\mathbf{Pr}[|y| < O(\sigma/\sqrt{\alpha})] \geq 1 - \alpha$, we see $\mathbf{Pr}[|x + y| < O(\sigma/\sqrt{\alpha})] \geq \alpha/2$. Hoeffding's inequality (Lemma A.1) now implies that the empirical median also satisfies the above upper bound as long as the number of samples is greater than $O(1)/\alpha^2 \log(1/\delta)$.

To see that this is tight, consider the distribution centered at 0, whose density function is $2/(y + 2)^3$ in the range $[1, \infty)$, and is 0 otherwise.

Call this $D_{y^{-3}}$. Observe that $\mathbf{Pr}_{D_{y^{-3}}}[z > t] < O(1) \int_t^\infty y^{-3}dy = C/t^2$.

Let $x$ be a symmetric distribution whose distribution takes the value 0 with probability $\alpha$ and takes the values $\pm\alpha^{-1/2}100C^{1/2} + 10$ with probability $0.5(1 - \alpha)$.

We show that the median of the distribution of $x + y$ where $y$ is drawn from $D_{y^{-3}}$, is larger than $\Omega(\alpha^{-1/2})$.

To see this, we show that the probability that $x + y$ takes a value smaller than $100\alpha^{-1/2}C^{1/2}$ is less than half, implying that the median has to be larger than this quantity.

$x$ takes three values. Note that $y + 100\alpha^{-1/2}C^{1/2} + 12$ places no mass in the region $(-\infty, 100\alpha^{-1/2}C^{1/2} + 10]$. So to estimate the probability that $x + y$ takes a value smaller than $100\alpha^{-1/2}C^{1/2} + 10$, we only need to consider contributions from the other two possible values. By

choosing $\alpha$ small enough, so that $100\alpha^{-1/2}C^{1/2} > 10$, we see

$$\mathbf{Pr}[x + y < 100\alpha^{-1/2}C^{1/2} + 10]$$
$$< 0.5(1-\alpha)\,\mathbf{Pr}[y < 200\alpha^{-1/2}C^{1/2} + 20] + \alpha\,\mathbf{Pr}[y < 100\alpha^{-1/2}C^{1/2} + 10]$$
$$< 0.5(1-\alpha)(1 - C/(200\alpha^{-1/2}C^{1/2} + 20)^2) + \alpha(1 - C/(100\alpha^{-1/2}C^{1/2} + 10)^2)$$
$$< 0.5(1-\alpha)(1 - \alpha^{1/2}/(400)^2) + \alpha$$
$$< 0.5 + 0.5\alpha - \alpha^{1/2}/8 \cdot (400^2) \, .$$

We are done when $0.5\alpha - \alpha^{1/2}/8 \cdot (400^2) < 0$, this happens for $\alpha^{1/2} < 2/(8 \cdot (400^2))$. $\qquad\square$

## B.2 Useful Lemma for Finer Estimation

In the second step of our location-estimation lemma, we refine the estimate of $t$. To do this, we first re-center the distributions based on our rough estimate, so that the shift after re-centering is bounded. Then, we identify an interval $I$ centered around 0 such that, when conditioning on $\xi + z$ falling within this interval, the expected value of $\xi + z$ remains the same as when conditioning on $\xi$ falling within the same interval. This expectation will help us get an improved estimate, which we use to get an improved re-centering of our original distributions, and repeat the process.

To identify such an interval, we search for a pair of bounded-length intervals equidistant from the origin (for e.g. $(-10\sigma, -5\sigma)$ and $(5\sigma, 10\sigma)$) that contain very little probability mass. By doing so, when $z$ is added to $\xi$, the amount of probability mass shifted into the interval $(-5\sigma, 5\sigma)$ $z$ remains small.

In this subsection, we prove Lemma B.2, which states that any positive sequence which has a finite sum must eventually have one small element. The lemma also gives a concrete upper bound on which element of the sequence satisfies this property.

**Lemma B.2.** $a_i \geq 0$ for all $i$ and $\sum_{i=0}^{\infty} a_i < C$ for some constant $C$. Also, suppose we have $\eta \in (0,1)$ and $L \in \mathbb{R}$ such that $L \geq 1$. Then there is an integer $i$ such that $L \leq i \leq (C/a_0 + L^\eta)^{1/\eta}$ and $ia_i < \eta \sum_{j=0}^{i} a_j$.

Consider a partition of the reals into length $L$ intervals. Here, $L$ simply denotes the length of the intervals and the lower bound to $i$, not the smoothness parameter.

In our proof, we will use Lemma B.2 on the sequence $a_i$, where $a_i$ corresponds to an upper bound on the mass of $\xi$ contained in the $i$-th intervals equidistant from the origin on either side, and the mass that crosses them (i.e., the mass of $\xi$ that is moved either inside or out of the interval when $z$ is added to it).

We need the following calculation to prove Lemma B.2.

Notation: For integer $i \geq 1$ and $\eta \in (0,1)$, define $(i - \eta)! := \Pi_{j=1}^{i}(j - \eta)$.

**Fact B.3.** *Let* $A_k := 1 + \sum_{t=1}^{k-1} \frac{\eta(t-1)!}{(t-\eta)!}$. *Then, for* $k \geq 2$, $A_k = (k-1)!/(k-1-\eta)!$.

*Proof.* We prove this by induction. By definition, our hypothesis holds for $A_2$ because $A_2 = 1 + \eta/(1-\eta) = 1/(1-\eta) = (2-1)!/(2-1-\eta)!$. Suppose it holds for all $2 \leq t \leq k$. We then show that it holds for $t = k + 1$.

$$A_{k+1} = 1 + \sum_{t=1}^{k} \frac{\eta(t-1)!}{(t-\eta)!} = A_k + \frac{\eta\,(k-1)!}{(k-\eta)!}$$
$$= \frac{(k-1)!}{(k-1-\eta)!} + \frac{\eta\,(k-1)!}{(k-\eta)!} = \frac{(k-1)!}{(k-1-\eta)!}\left(1 + \frac{\eta}{k-\eta}\right)$$
$$= \frac{(k-1)!}{(k-1-\eta)!}\frac{k}{k-\eta} = \frac{k!}{(k-\eta)!}.$$

$\qquad\square$

*Proof of Lemma B.2* Let $U = (C/a_0 + L^\eta)^{1/\eta}$ and suppose towards a contradiction that there is no such $i$ that satisfies the lemma. That means, we will assume that $ia_i \geq \eta \sum_{j=0}^{i} a_j$ for all integers $i \in [1, U]$; or equivalently, we assume $a_i \geq \frac{\eta}{i-\eta} \sum_{j=0}^{i-1} a_j$. We then show that this implies $i^{1-\eta} a_i \geq \eta a_0$ for all $i$ in the range.

Consider the inductive hypothesis on $t$ given by $a_t \geq \eta \frac{(t-1)!}{(t-\eta)!} \cdot a_0$. The base case when $t = 1$ is true since $a_1 \geq \eta a_0/(1-\eta)$ by our assumption. Suppose the inductive hypothesis holds for integers $t \in [1, k-1]$. We show this for $t = k$ below. By starting with our assumption then applying the inductive hypothesis, we have that

$$
\begin{aligned}
a_k &\geq \frac{\eta}{k-\eta} \sum_{t=0}^{k-1} a_t \\
&\geq \frac{a_0 \eta}{k-\eta} \left( 1 + \sum_{t=1}^{k-1} \frac{\eta(t-1)!}{(t-\eta)!} \right) \\
&= a_0 \eta \frac{(k-1)!}{(k-\eta)!} \; .
\end{aligned}
$$

The final equality follows from Fact B.3 which states that $(k-1)!/(k-1-\eta)! = 1 + \sum_{t=1}^{k-1} \frac{\eta(t-1)!}{(t-\eta)!}$. Simplifying this further, we see that since $(i - \eta) \geq i \exp(-\eta/i)$ for all $i \in [1, k]$,

$$
\begin{aligned}
a_k &\geq a_0 \eta \frac{(k-1)!}{(k-\eta)!} \\
&\geq a_0 \eta \frac{(k-1)!}{k! \exp(-\eta/k)} \\
&\geq a_0 \eta \, (1/k) \, (1/\exp(-\eta(\sum_{i=1}^{k} 1/i))) \\
&\geq a_0 \eta \, (1/k) \, (1/\exp(-\eta \log(k)/20)) \\
&\geq (a_0/2)\eta \, (1/k^{1-\eta/20}) \; .
\end{aligned}
$$

Finally, observe that

$$
\begin{aligned}
C > \sum_{i=L}^{U} a_i &> a_0 \eta \sum_{i=L}^{U} (1/i^{1-\eta/20}) \\
&> a_0 \eta \int_{L}^{U} (1/x^{1-\eta}) \, dx \\
&= a_0(U^\eta - L^\eta).
\end{aligned}
$$

By definition, $U = (C/a_0 + L^\eta)^{1/\eta}$ so the above inequality cannot be true and thus we have arrived at a contradiction. Then it follows that there exists an $i$ such that $1 \leq L \leq i \leq (C/a_0 + L^\eta)^{1/\eta}$ and $ia_i < \eta \sum_{j=0}^{i} a_j$. $\qquad\square$

## C  Noisy Location Estimation

In this section, we state and prove the guarantees of our algorithms for noisy location estimation (Lemma 3.1 and Lemma 3.5).

### C.1  One-dimensional Noisy Location Estimation

Throughout the technical summary and some parts of the proof, we make the assumption that the variables $y$ and $y'$ were bounded. Extending this assumption to bounded-variance distributions requires significant effort.

Our algorithm for one-dimensional noisy location estimation (Algorithm 1) can be thought of as a two-step process. The first step involves a rough initial estimation algorithm, while the second step employs an iterative algorithm that progressively refines the estimate by a factor of $\eta$ in each iteration.

Due to space limitations and for ease of exposition, the algorithm we present in the main body is a sketch of the refinement procedure.

In this Algorithm 1 , we introduce the definition of $\hat{P}$ (the empirical estimate of $\tilde{P}(\cdot)$), which is an upper bound on the probability mentioned earlier. This probability can be calculated using samples from $\xi + z$ and $\xi + z'$. Additionally, we incorporate the iterative refinement process within the algorithm.

---

**Algorithm 4** One-dimensional Location Estimation: $\text{Shift1D}(S_1, S_2, \eta, \sigma, \alpha)$

---

**Input:** Sample sets $S_1, S_2 \subset \mathbb{R}^d$ of size $m$, $\alpha, \eta \in (0, 1)$, $\sigma > 0$

1. Let $T = O(\log_{1/\eta}(1/\alpha))$. For $j \in \{1, 2\}$, partition $S_j$ into $T$ equal pieces, $S_j^{(i)}$ for $i \in [T]$.
2. $D = \{a - b \mid a \in S_1^{(1)}, b \in S_2^{(1)}\}$.
3. $t'(1) := \text{median}(D)$.
4. Set $A = O(1/\sqrt{\alpha})$.
5. Repeat steps 6 to 12, for $i$ going from 2 to $T$:
6. $S_1^{(i)} := S_1^{(i)} - t'_r(i - 1)$.
7. For $j \in \{1, 2\}$

$$\hat{P}_j(i) := O(1) \Pr_{x \sim S_j^{(i)}}[|x| \in A\sigma(i - 5, i + 5)]$$

$$+ O(1) \sum_{j=1}^{i-1} (1/(i - j)^2) \Pr_{x \sim S_j^{(i)}}[|x| \in Aj\sigma + A\sigma[-4, 5).]$$

8. Let $\hat{P}(i) = \hat{P}_1(i) + \hat{P}_2(i)$.
9. Identify an integer $k \in [(1/\alpha\eta), (C/\alpha + 1/(\alpha\eta)^\eta)^{1/\eta}]$ such that

$$\hat{P}(k) \leq \eta \sum_{j \in \{1, 2\}} \Pr_{x \sim S_j^{(i)}}[|x| \in A\sigma k] \pm O(\eta/i).$$

10. $t'(i) := t'(i - 1) + \mathbf{E}_{z \sim S_1^{(i)}}[z \mid |z| \leq A\sigma k] - \mathbf{E}_{z \sim S_2^{(i)}}[z \mid |z| \leq A\sigma k]$.
11. $A := \eta A$.
12. Return $t'(T)$

---

**Lemma C.1** (One-dimensional location-estimation). *There is an algorithm (Algorithm 1) which, given $(1/\eta^5)(O(1)/\alpha)^{2/\eta} \log(1/\eta\alpha\delta)$ samples $\{\xi_i + y_i + t\}_{i=1}^m$ and $\{\tilde{\xi}_i + y'_i\}_{i=1}^m$ where $\xi_i$ and $\tilde{\xi}_i$ are both drawn from $D_\xi$, $y_i$ and $y'_i$ are drawn from distinct distributions with mean-zero with variance bounded above by $\sigma$ and $t \in \mathbb{R}$ is an unknown translation, runs in time $\tilde{\text{poly}}(1/\eta, (O(1)/\alpha)^{1/\eta}, \log(1/\delta\eta\alpha))$ and recovers $t'$ such that $|t - t'| \leq O(\eta\sigma)$.*

*Proof.* Our proof is based on the following claims:

**Claim C.2** (Rough Estimate). *There is an algorithm which, given $m = O((1/\alpha^4) \log(1/\delta))$ samples of the kind $\xi + y + t$ and $\xi + y'$, where $t \in \mathbb{R}$ is an unknown translation, returns $t'_r$ satisfying $|t'_r - t| < O(\sigma\alpha^{-1/2})$.*

**Claim C.3** (Fine Estimate). *Suppose $z, z'$ have means bounded from above by $A\sigma$ and variances at most $\sigma^2$ and suppose $\alpha \in (0, 1)$ and $\eta \in (0, 1/2)$. Then in $\text{poly}((O(1)/\alpha\eta)^{1/\eta}, \log(1/\delta\eta\alpha))$ samples and $\text{poly}((O(1)/\alpha\eta)^{1/\eta}, \log(1/\delta\eta\alpha))$ time, it is possible to recover $k \in [1/\eta\alpha, (O(1)/\eta\alpha)^{O(1/\eta)}]$ such that*

$$\widehat{\mathbf{E}}[\xi + z \mid |\xi + z| \leq A\sigma k] - \widehat{\mathbf{E}}[\xi + z' \mid |\xi + z'| \leq A\sigma k] = \mathbf{E}[z] - \mathbf{E}[z'] \pm \eta\,(A\sigma).$$

Using Claim 3.2, we first identify a rough estimate $t'_r$ satisfying $|t'_r - t| < O(\sigma\alpha^{-1/2})$. This allows us to re-center $y'$. Let the re-centered distribution be denoted by $z' = y'$ and $z = y + t - t'_r$. Then $z$ and $z'$ are such that $\mathbf{E}[z]$ and $\mathbf{E}[z']$ are both at most $O(\sigma\alpha^{-1/2})$ in magnitude, and have variance at most $\sigma^2$.

Claim 3.4 then allows us to estimate $t'_f$ such that $|(\mathbf{E}[z] - \mathbf{E}[z']) - t'_f| = |t - t'_r - t'_f| \leq \eta\, O(\sigma\alpha^{-1/2})$.

Setting $t' = t'_r + t'_f$, we see that our estimate $t'$ is now $\eta$ times closer to $t$ compared to $t'_r$.

To refine this estimate further, we can obtain fresh samples and re-center using $t'$ instead of $t'_r$. Repeating this process $O(\log_{1/\eta}(1/\alpha)) = O(\log_\eta(\alpha))$ times is sufficient to obtain an estimate that incurs an error of $\eta \cdot \eta^{\log_\eta(\alpha^{1/2})} \cdot O(\sigma\alpha^{-1/2}) \leq O(\eta\sigma)$.

This results in a runtime and sample complexity that is only $O(\log_{1/\eta}(1/\alpha))$ times the runtime and sample complexity required by Claim 3.4. This amounts to the final runtime and sample complexity being $\mathrm{poly}((O(1)/\alpha\eta)^{1/\eta}, \log(1/\delta\eta\alpha))$.

We now prove Claim 3.2 and Claim 3.4.

Claim 3.2 shows that the median of the distribution of pairwise differences of $\xi + y + t$ and $\xi + y'$ estimates the mean up to an error of $\sigma\alpha^{-1/2}$.

*Proof of Claim 3.2*   Let $\tilde{\xi}$ be a random variable with the same distribution as $\xi$ and independently drawn. We have independent samples from the distributions of $\xi + y + t$ and $\xi + y'$. Applying Fact B.1 to these distributions, we see that if we have at least $O(1/\alpha^4)\log(1/\delta)$ samples from the distribution of $(\xi - \tilde{\xi}) + (y - y') + t$, these samples will have a median of $t \pm O(\sigma/\sqrt{\alpha})$.   $\square$

*Proof of Claim 3.4*   To identify such a $k$, the idea is to ensure that $\mathbf{E}[\xi + z \mid |\xi + z| \leq A\sigma k] = \mathbf{E}[\xi + z \mid |\xi| \leq A\sigma k] \pm O(A\eta\sigma) = \mathbf{E}[\xi \mid |\xi| \leq A\sigma k] + \mathbf{E}[z] \pm O(A\eta\sigma)$, and similarly for $z'$. The theorem follows by taking the difference of these equations.

Before we proceed, we will need the following definitions: let $P(i, z)$ be defined as follows:

$$P(i, z) := \mathbf{Pr}[|\xi| \in A\sigma(i - 1, i + 1)]$$
$$+ \mathbf{Pr}[|\xi| < Ai\sigma, |\xi + z| > Ai\sigma] + \mathbf{Pr}[|\xi| > Ai\sigma, |\xi + z| < Ai\sigma]\,.$$

This will help us bound the final error terms that arise in the calculation. We will need the following upper bound on $P(i, z) + P(i, z')$.

**Claim C.4.** *There exists a function $\tilde{P} : \mathbb{N} \to \mathbb{R}^+$ satisfying:*

1. *For all $i \in \mathbb{N}$, $\tilde{P}(i) \geq P(i, z) + P(i, z')$ which can be computed using samples from $\xi + z$ and $\xi + z'$.*

2. *There is a $k \in [(1/\alpha\eta), (C/\alpha + 1/(\alpha\eta)^\eta)^{1/\eta}]$ such that $k\tilde{P}(k) < \eta \sum_{j=0}^k \tilde{P}(k)$.*

3. *$\sum_{j=0}^k \tilde{P}(k) = O(\mathbf{Pr}[|\xi + z| \leq A\sigma k] + \mathbf{Pr}[|\xi + z'| \leq A\sigma k])$.*

4. *With probability $1 - \delta$, for all $i < (O(1)/\alpha)^{1/\eta}/\eta$, $\tilde{P}(i)$ can be estimated to an accuracy of $O(\eta/i)$ by using $(O(1)/\alpha)^{2/\eta} \log(1/\eta\alpha\delta)/\eta^5$ samples from $\xi + z$ and $\xi + z'$.*

We defer the proof of Claim C.4 to Appendix C.2, and continue with our proof showing that $\mathbf{E}[\xi + z \mid |\xi + z| \leq A\sigma k] \approx \mathbf{E}[\xi + z \mid |\xi| \leq A\sigma k]$ for $k$ satisfying the conclusions of Claim C.4. To this end, observe the following for $f(\xi, z)$ being either 1 or $\xi + z$.

$$|\mathbf{E}[f(\xi, z)\,\mathbf{1}(|\xi| \leq \sigma i)] - \mathbf{E}[f(\xi, z)\,\mathbf{1}(|\xi + z| \leq \sigma i)]|$$
$$\leq |\mathbf{E}[f(\xi, z)\,\mathbf{1}(|\xi + z| > \sigma i)\,\mathbf{1}(|\xi| \leq \sigma i)]| + |\mathbf{E}[f(\xi, z)\,\mathbf{1}(|\xi + z| \leq \sigma i)\,\mathbf{1}(|\xi| > \sigma i)]|.$$

By setting $f(\xi, z) := 1$ and considering the case where $i = k$ satisfies the conclusions of Claim C.4, we can bound the "error terms"

$$\mathbf{Pr}[|\xi + z| \leq A\sigma k \text{ and } |\xi| > A\sigma k] \text{ and } \mathbf{Pr}[|\xi + z| > A\sigma k \text{ and } |\xi| \leq A\sigma k] \text{ in terms of } \tilde{P}(k).$$

Furthermore, $\tilde{P}(k)$ itself is upper bounded by $O(\eta/k)(\mathbf{Pr}[|\xi + z| \leq A\sigma k] + \mathbf{Pr}[|\xi + z'| \leq A\sigma k])$ as per Item 2 and Item 3. Putting these facts together, we have that

$$|\mathbf{Pr}[|\xi| \leq A\sigma k] - \mathbf{Pr}[|\xi + z| \leq A\sigma k]|$$
$$= O(\eta/k)(\mathbf{Pr}[|\xi + z| \leq A\sigma k] + \mathbf{Pr}[|\xi + z'| \leq A\sigma k]).$$

A similar claim holds for the distribution over $z'$. An application of the triangle inequality now implies

$$|\mathbf{Pr}[|\xi + z'| \leq A\sigma k] - \mathbf{Pr}[|\xi + z| \leq A\sigma k]|$$
$$= O(\eta/k)\left(\mathbf{Pr}[|\xi + z| \leq A\sigma k] + \mathbf{Pr}[|\xi + z'| \leq A\sigma k]\right).$$

If $|A - B| < \tau(A + B)$ it follows that $(1 - \tau)/(1 + \tau) < A/B < (1 + \tau)/(1 - \tau)$. For $\tau \in (0, 1/2]$, this means $A = \Theta(B)$. Applying this to our case, we can conclude that $\mathbf{Pr}[|\xi + z'| \leq A\sigma k] = \Theta(\mathbf{Pr}[|\xi + z| \leq A\sigma k])$. Substituting this equivalence back into the previous expression, we obtain:

$$|\mathbf{Pr}[|\xi| \leq A\sigma k] - \mathbf{Pr}[|\xi + z| \leq A\sigma k]| = O(\eta/k)\left(\mathbf{Pr}[|\xi + z| \leq A\sigma k]\right). \qquad (1)$$

Similarly, when $f(\xi, z) := \xi + z$, we need to control the error terms: $\mathbf{E}[(\xi + z)\,\mathbf{1}(|\xi + z| \leq A\sigma k)\,\mathbf{1}(|\xi| > A\sigma k)]$ and $\mathbf{E}[(\xi + z)\,\mathbf{1}(|\xi + z| > A\sigma k)\,\mathbf{1}(|\xi| \leq A\sigma k)]$.

Observe that $(\xi + z)\,\mathbf{1}(|\xi + z| \leq A\sigma k)\,\mathbf{1}(|\xi| > A\sigma k)$ has a nonzero value with probability at most $\mathbf{E}[\mathbf{1}(|\xi + z| \leq A\sigma k)\,\mathbf{1}(|\xi| > A\sigma k)] < \tilde{P}(k)$. Also, the magnitude of $(\xi + z)$ in this event is at most $A\sigma k$. Putting these together, we get that

$$|\mathbf{E}[(\xi + z)\,\mathbf{1}(|\xi + z| \leq A\sigma k)\,\mathbf{1}(|\xi| > A\sigma k)]| < A\sigma k \tilde{P}(k) < O(A\sigma\eta)\sum_{j=1}^{k}\tilde{P}(j).$$

Unfortunately, we cannot use the same argument to bound $\mathbf{E}[(\xi + z)\,\mathbf{1}(|\xi + z| > A\sigma k)\,\mathbf{1}(|\xi| \leq A\sigma k)]$, since $|\xi + z|$ is no longer bounded by $A\sigma k$ in this event. However, we can break the sum $\xi + z$ as follows: $\xi + z = \xi + z\mathbf{1}(|z| > A\sigma k) + z\mathbf{1}(|z| \leq A\sigma k)$. This allows us to get the following bound:

$$\mathbf{E}[(\xi + z)\,\mathbf{1}(|\xi + z| > A\sigma k)\,\mathbf{1}(|\xi| \leq A\sigma k)]$$
$$< 2A\sigma k\tilde{P}(k) + \mathbf{E}[z\,\mathbf{1}(|z| > A\sigma k)\,\mathbf{1}(|\xi + z| > A\sigma k)\,\mathbf{1}(|\xi| \leq A\sigma k)]$$
$$< 2A\sigma k\tilde{P}(k) + \mathbf{E}[z\,\mathbf{1}(|z| > A\sigma k)]$$
$$< 2A\sigma k\tilde{P}(k) + A\sigma/k$$
$$< O(A\eta\sigma\sum_{j=1}^{k}\tilde{P}(j)) + O(A\eta\sigma\alpha),$$

where the third inequality follows by an application of Chebyshev's inequality, and the final inequality follows by choosing $k \geq 1/(\eta\alpha)$.

Putting everything together, we see

$$\mathbf{E}[(\xi + z)\,\mathbf{1}(|\xi + z| \leq A\sigma k)] = \mathbf{E}[(\xi + z)\,\mathbf{1}(|\xi| \leq A\sigma k)] \pm O(A\sigma\eta\alpha + A\sigma\eta\sum_{j=1}^{k}\tilde{P}(j)). \qquad (2)$$

To finally compute the conditional probability, we use Equation (1) and Equation (2) to get

$$\mathbf{E}[(\xi + z) \mid |\xi| \leq A\sigma k] = \frac{\mathbf{E}[(\xi + z)\,\mathbf{1}(|\xi + z| \leq A\sigma k)] \pm O(A\sigma\eta\alpha + A\sigma\eta\sum_{j=1}^{k}\tilde{P}(j))}{\mathbf{Pr}[|\xi| \leq A\sigma k]}$$
$$= \frac{\mathbf{E}[(\xi + z)\,\mathbf{1}(|\xi + z| \leq A\sigma k)] \pm O(A\sigma\eta\alpha + A\sigma\eta\sum_{j=1}^{k}\tilde{P}(j))}{(1 + \Theta(\eta/k))\,\mathbf{Pr}[|\xi + z| \leq A\sigma k]}$$
$$= (1 - \Theta(\eta/k))\,\mathbf{E}[(\xi + z) \mid |\xi + z| \leq A\sigma k]$$
$$\quad \pm O(1)\frac{A\sigma\eta\alpha + A\sigma\eta\,\mathbf{Pr}[|\xi + z| \leq A\sigma k]}{\mathbf{Pr}[|\xi + z| \leq A\sigma k]}$$
$$= \mathbf{E}[(\xi + z) \mid |\xi + z| \leq A\sigma k] \pm O(A\eta\sigma),$$

where the second inequality is a consequence of Item 3, and the last is due to the fact that $\mathbf{Pr}[|\xi+z| \leq A\sigma k] \geq \alpha/2$ whenever $k > 2$, which follows from an application of Fact A.3 while noting the fact that $\mathbf{Pr}[\xi = 0] \geq \alpha$.

Taking a difference for the above calculations for $z$ and $z'$, we see that,

$$\mathbf{E}[(\xi + z) \mid |\xi + z| \leq A\sigma k] - \mathbf{E}[(\xi + z') \mid |\xi + z'| \leq A\sigma k] = \mathbf{E}[z] - \mathbf{E}[z'] \pm O(A\eta\sigma).$$

Consider this final error, and let $O(A\eta\sigma) < CA\eta\sigma$ for some constant $C$. Repeating the above argument initially setting $\eta = \eta'/C$, where $C$ is the constant gives us the guarantee we need.

Finally, we estimate the runtime and sample complexity of our algorithm. The main bottleneck in our algorithm is the repeated estimation of $\tilde{P}(i)$ and estimation of $\mathbf{E}[(\xi + z) \mid |\xi + z| \leq A\sigma k]$.

According to Item 4, each time we estimate $\tilde{P}(i)$ to the desired accuracy, we draw $\mathrm{poly}(1/\eta, (O(1)/\alpha)^{1/\eta}, \log(1/\delta\eta\alpha))$ samples.

An application of Hoeffding's inequality (Lemma A.1) then allows us to estimate the conditional expectation $\mathbf{E}[(\xi + z) \mid |\xi + z| \leq A\sigma k]$ to an accuracy of $\eta A\sigma$ by drawing $\mathrm{poly}(1/\eta, (O(1)/\alpha)^{1/\eta}, \log(1/\delta\eta\alpha))$ samples as well. The exponential dependence here comes from the exponential upper bound on $k$.

$\square$

## C.2  Proof of Claim C.4

In this section, we prove the existence of $\tilde{P}(\cdot)$ which is an upper bound on $P(i, z) + P(i, z')$, which we can estimate using samples from $\xi + z$ and $\xi + z'$.

*Proof of Claim C.4*

*Proof of Item 1*:

Recall the definition of $P(i, z)$.

$$P(i, z) := \mathbf{Pr}[|\xi| \in A\sigma(i-1, i+1)]$$
$$+ \mathbf{Pr}[|\xi| < Ai\sigma, |\xi + z| > Ai\sigma] + \mathbf{Pr}[|\xi| > Ai\sigma, |\xi + z| < Ai\sigma].$$

For Item 1 to hold, we need to define $\tilde{P}(i)$ to be an upper bound on $P(i, z) + P(i, z')$ which can be computed using samples from $\xi + z$ and $\xi + z'$. To this end, we bound $P(i, z)$ as follows. First, note that we can adjust the endpoints of the intervals to get

$$P(i, z) < 3\,\mathbf{Pr}[|\xi| \in A\sigma(i-1, i+1)]$$
$$+ \mathbf{Pr}[|\xi| < A(i-1)\sigma, |\xi + z| > Ai\sigma] + \mathbf{Pr}[|\xi| > A(i+1)\sigma, |\xi + z| < Ai\sigma].$$

Then, we partition the ranges in the definition above into intervals of length $A\sigma$ to get:

$$P(i, z) < 3\,\mathbf{Pr}[|\xi| \in A\sigma(i-1, i+1)]$$
$$+ \sum_{j=1}^{i-2} \mathbf{Pr}[|\xi| \in Aj\sigma + [0, A\sigma), |\xi + z| > Ai\sigma]$$
$$+ \sum_{j=1}^{i-1} \mathbf{Pr}[|\xi| > A(i+1)\sigma, |\xi + z| \in Aj\sigma + [0, A\sigma)].$$

Next, an application of the triangle inequality to $|\xi| \in Aj\sigma + [0, A\sigma)$ and $|\xi + z| > Ai\sigma$ implies that $|z| \geq A(i - j - 1)\sigma$. Similarly, the same kind of argument when $|\xi| > A(i + 1)\sigma$ and $|\xi + z| \in Aj\sigma + [0, A\sigma)$ demonstrates that $|-z| = |\xi + z - \xi| \geq A(i - j)\sigma$. We then use Fact A.3 to move from $|\xi + z|$ to $|\xi|$ in the third term.

$$P(i, z) < 3\,\mathbf{Pr}[|\xi| \in A\sigma(i-1, i+1)]$$
$$+ \sum_{j=1}^{i-2} \mathbf{Pr}[|\xi| \in Aj\sigma + A\sigma[0, 1), |z| \geq (i - j - 1)A\sigma]$$
$$+ O(1) \sum_{j=1}^{i-1} \mathbf{Pr}[|\xi| \in Aj\sigma + A\sigma[-2, 3), |z| \geq (i - j)A\sigma].$$

An application of Chebyshev's inequality to $z$, using the independence of $z$ and $\xi$, gives that

$$P(i,z) < 3\,\mathbf{Pr}[|\xi| \in A\sigma(i-1, i+1)]$$
$$+ O(1)\sum_{j=1}^{i-2}(1/(i-j-1)^2)\,\mathbf{Pr}[|\xi| \in Aj\sigma + A\sigma[0,1]]$$
$$+ O(1)\sum_{j=1}^{i-1}(1/(i-j)^2)\,\mathbf{Pr}[|\xi| \in Aj\sigma + A\sigma[-2,3]]\,.$$

Another application of Fact A.3 applied to $(\xi + z) - z$ then gives us

$$P(i,z) < 3\,\mathbf{Pr}[|\xi + z| \in A\sigma(i-5, i+5)]$$
$$+ O(1)\sum_{j=1}^{i-2}(1/(i-j-1)^2)\,\mathbf{Pr}[|\xi + z| \in Aj\sigma + A\sigma[-2,3]]$$
$$+ O(1)\sum_{j=1}^{i-1}(1/(i-j)^2)\,\mathbf{Pr}[|\xi + z| \in Aj\sigma + A\sigma[-4,5]].$$

Finally, extending all intervals so that they match, and observing that $\sum_{j=1}^{i-2}(1/(i-j-1)^2)\,\mathbf{Pr}[|\xi + z| \in Aj\sigma + A\sigma[-2,3]] \le \sum_{j=1}^{i-1}(1/(i-j)^2)\,\mathbf{Pr}[|\xi + z| < Aj\sigma + A\sigma[-4,5]]$, we get

$$P(i,z) < O(1)\,\mathbf{Pr}[|\xi + z| \in A\sigma(i-5, i+5)]$$
$$+ O(1)\sum_{j=1}^{i-1}(1/(i-j)^2)\,\mathbf{Pr}[|\xi + z| \in Aj\sigma + A\sigma[-4,5]].$$

We now let $\tilde{P}(i,z)$ denote the final upper bound on $P(i,z)$. The value of having $\tilde{P}(i,z)$ is that it can be computed using samples from $\xi + z$.

$$\tilde{P}(i,z) := O(1)\,\mathbf{Pr}[|\xi + z| \in A\sigma(i-5, i+5)]$$
$$+ O(1)\sum_{j=1}^{i-1}(1/(i-j)^2)\,\mathbf{Pr}[|\xi + z| \in Aj\sigma + A\sigma[-4,5]]\,.$$

We defined $\tilde{P}(i) = \tilde{P}(i,z) + \tilde{P}(i,z')$.

*Proof of Item 2:*

First observe that $\sum_{i=0}^{\infty}\tilde{P}(i) < C$ for some constant $C$. It is clear that this is true of the first term, since every interval will get over-counted at most 10 times. To see that the second term can be bounded, observe that

$$\sum_{i=1}^{\infty}\sum_{j=1}^{i-1}(1/(i-j)^2)\,\mathbf{Pr}[|\xi + z| \in Aj\sigma + A\sigma[-4,5]]$$
$$< \sum_{j=1}^{\infty}\sum_{i=1}^{\infty}(1/(i-j)^2)\,\mathbf{Pr}[|\xi + z| \in Aj\sigma + A\sigma[-4,5]]$$
$$< \sum_{j=1}^{\infty}\mathbf{Pr}[|\xi + z| \in Aj\sigma + A\sigma[-4,5]]\sum_{i=1}^{\infty}(1/(i-j)^2) = O(1)\,.$$

The first inequality follows by extending the limits of summation.

The final inequality follows from the fact that the total probability is at most 1, every interval of size $\sigma$ gets over-counted at most finitely many times, and the fact that $\sum_{k=1}^{\infty}1/k^2 = O(1)$.

Item 2 now follows from the fact that $\tilde{P}(i)$, $i \ge 0$ is a non-negative sequence that sums to a finite quantity, and $\tilde{P}(0) \ge \alpha/2$, since the interval $\tilde{P}(0)$ upper bounds is contains at least a constant fraction of the mass of $\xi$ at 0 that is moved by $z, z'$, and $\mathbf{Pr}[\xi = 0] \ge \alpha$.

Applying Lemma B.2, we get our result.

*Proof of Item 3:*

Let $k$ be such that Item 2 holds, i.e. $k\tilde{P}(k) < \eta \sum_{j=1}^{k} \tilde{P}(k)$, then the goal is to show $\sum_{j=1}^{k} \tilde{P}(k) = O(\mathbf{Pr}[|\xi + z| < A\sigma k] + \mathbf{Pr}[|\xi + z'| < A\sigma k])$.

We first consider the sum over $i$, of $\tilde{P}(i, z)$. It is easy to see that this is

$$\sum_{i=1}^{k} \tilde{P}(i, z) = O(1)\,\mathbf{Pr}[|\xi + z| \le A\sigma(k+5)]$$

$$+ O(1) \sum_{i=1}^{k} \sum_{j=1}^{i-1} (1/(i-j)^2)\,\mathbf{Pr}[|\xi + z| \in Aj\sigma + A\sigma[-4, 5]]\,.$$

The first term on the RHS is almost what we want. We now show how to bound the second term,

$$\sum_{i=1}^{k} \sum_{j=1}^{i-1} (1/(i-j)^2)\,\mathbf{Pr}[|\xi + z| \in Aj\sigma + A\sigma[-4, 5]]$$

$$< \sum_{j=1}^{k-1} \sum_{i=0; i\ne j}^{k} (1/(i-j)^2)\,\mathbf{Pr}[|\xi + z| \in Aj\sigma + A\sigma[-4, 5]]$$

$$= \sum_{j=1}^{k-1} \mathbf{Pr}[|\xi + z| \in Aj\sigma + A\sigma[-4, 5]] \sum_{i=0; i\ne j}^{k} (1/(i-j)^2)$$

$$< O(1) \sum_{j=1}^{k-1} \mathbf{Pr}[|\xi + z| \in Aj\sigma + A\sigma[-4, 5]]$$

$$< O(1)\,\mathbf{Pr}[|\xi + z| < A(k+5)\sigma]\,.$$

The first inequality holds since any pair of $(i, j)$ that has a nonzero term in the first sum will also occur in the second sum, and all terms are non-negative.

The second equality is just pulling the common $j$ term out.

The third inequality follows from the fact that $\sum_{i=1}^{\infty} 1/i^2 = O(1)$.

The fourth inequality follows from the fact that each $\sigma$-length interval is overcounted at most a constant number of times.

This allows us to bound $\sum_{i=1}^{k} \tilde{P}(i, z)$ by $O(\mathbf{Pr}[|\xi + z| < A\sigma(k+5)])$ overall. Similarly for $\sum_{i=1}^{k} \tilde{P}(i, z')$, we can obtain a bound of $O(\mathbf{Pr}[|\xi + z'| < A\sigma(k+5)])$. Putting these together, we see $\sum_{i=1}^{k} \tilde{P}(i) \le O(\mathbf{Pr}[|\xi + z| < A\sigma(k+5)] + \mathbf{Pr}[|\xi + z'| < A\sigma(k+5)])$. Finally, to get the upper bound claimed in Item 3, observe that

$$\mathbf{Pr}[|\xi + z| < A\sigma(k+5)] + \mathbf{Pr}[|\xi + z'| < A\sigma(k+5)]$$

$$= \mathbf{Pr}[|\xi + z| < A\sigma(k-4)] + \mathbf{Pr}[|\xi + z'| < A\sigma(k-4)]$$

$$\quad + \mathbf{Pr}[|\xi + z| \in A\sigma(k-4, k+5)] + \mathbf{Pr}[|\xi + z'| \in A\sigma(k-4, k+5)]$$

$$\le \mathbf{Pr}[|\xi + z| < A\sigma(k-4)] + \mathbf{Pr}[|\xi + z'| < A\sigma(k-4)]$$

$$\quad + \tilde{P}(k)$$

$$\le \mathbf{Pr}[|\xi + z| < A\sigma(k-4)] + \mathbf{Pr}[|\xi + z'| < A\sigma(k-4)]$$

$$\quad + O(\eta/k)(\mathbf{Pr}[|\xi + z| < A\sigma(k+5)] + \mathbf{Pr}[|\xi + z'| < A\sigma(k+5)])\,.$$

Rearranging the inequality and by scaling $\eta$ such that that $O(\eta/k) \le 1/2$, we see that $\mathbf{Pr}[|\xi + z| < A\sigma(k+5)] + \mathbf{Pr}[|\xi + z'| < A\sigma(k+5)] = O(\mathbf{Pr}[|\xi + z| < A\sigma(k-4)] + \mathbf{Pr}[|\xi + z'| < A\sigma(k-4)]) = O(\mathbf{Pr}[|\xi + z| < A\sigma k] + \mathbf{Pr}[|\xi + z'| < A\sigma k])$, completing our proof of Item 3.

*Proof of Item 4:*

Finally, to see Item 4 holds, observe that $0 < \tilde{P}(i) < O(1)$. Let $B = (1/\eta)(O(1)/\alpha)^{1/\eta}$ denote the maximum index before which we can find a $k$ such that $k\tilde{P}(k) \le \eta \sum_{i=0}^{k} \tilde{P}(i)$. Now, To estimate $\tilde{P}(i)$ empirically, we partition the interval $(-BA\sigma, BA\sigma)$ into $B$ intervals of length $A\sigma$ each, and estimate the probability of $\xi + z$ falling in each interval. If we estimate each of these probabilities to an accuracy of $\eta/(100\,B)$, we can estimate $\tilde{P}(i)$ to an accuracy of $O(\eta/i)$.

An application of Hoeffding's inequality (Lemma A.1) tells us that each estimate will require $O(B^2/\eta^2 \log(1/\delta))$ samples. Taking a union bound over all these intervals, we see that we will require $O(B^2/\eta^2 \log(B/\delta))$ samples.

Finally, another union bound over each $i \in [0, B]$ implies that we will need $O(B^2/\eta^2 \log(B^2/\delta))$ samples. Substituting the value of $B$ back in, we see that this amounts to requiring $(1/\eta^5)(O(1)/\alpha)^{2/\eta} \log(1/\eta\alpha\delta)$ samples.

Estimating $\tilde{P}(i)$ will take time polynomial in the number of samples, and so we take time $\mathrm{poly}((O(1)/\alpha)^{1/\eta}, 1/\eta, \log(1/\eta\alpha\delta))$. $\hspace{1cm}\square$

### C.3 High-dimensional Noisy Location Estimation

In this section, we explain how to use our one-dimensional location estimation algorithm to get an algorithm for noisy location estimation in $d$ dimensions.

The algorithm performs one-dimensional location estimation coordinate-wise, after a random rotation.

We need to perform such a rotation to ensure that every coordinate has a known variance bound of $\sigma/\sqrt{d}$.

---

**Algorithm 5** High-dimensional Location Estimation: ShiftHighD$(S_1, S_2, \eta, \sigma, \alpha)$

---

**input:** Sample sets $S_1, S_2 \subset \mathbb{R}^d$ of size $m$, $\eta \in (0, 1)$, $\sigma > 0$, $\alpha$

1. Sample $R_{i,j}$ i.i.d. from the uniform distribution over $\{\pm 1/\sqrt{d}\}$ for $i, j \in [d]$
2. Represent $S_1$ and $S_2$ in the basis given by the rows of $R$: $r_1, \ldots, r_d$.
3. **for** $i \in [d]$ **do**
$\quad\mid\quad v_i' := \mathrm{Shift1D}(S_1 \cdot e_i, S_2 \cdot e_i, \eta, O(\sigma/\sqrt{d}), \alpha)$
**end**
4. Change the representation of $v'$ back to the standard basis.
5. *Probability Amplification:* Repeat steps 1-4, $T := O(\log(1/\delta))$ times to get $C := \{v_1', \ldots, v_T'\}$
6. Find a ball of radius $O(\eta\sigma)$ centered at one of the $v_i'$ containing $> 90\%$ of $C$. If such a vector exists, set $v'$ to be this vector. Otherwise set $v'$ to be an arbitrary element of $C$.
5. Return $v'$.

---

**Lemma C.5** (Location Estimation). *Let $D_{y_i}$ for $i \in \{1, 2\}$ be the distributions of $\xi + z_i$ for $i \in \{1, 2\}$ where $\xi$ is drawn from $D_\xi$ and $\mathbf{Pr}_{\xi \sim D_\xi}[\xi = 0] \ge \alpha$ and $z_i \sim D_i$ are distributions over $\mathbb{R}^d$ satisfying $\mathbf{E}_{D_i}[x] = 0$ and $\mathbf{E}_{D_i}[\|x\|^2] \le \sigma^2$. Let $v \in \mathbb{R}^d$ be an unknown shift, and $D_{y_2,v}$ denote the distribution of $y_2$ shifted by $v$. There is an algorithm (Algorithm 5), which draws $\mathrm{poly}(1/\eta, (O(1)/\alpha)^{1/\eta}, \log(1/\delta\eta\alpha))$ samples each from $D_{y_1}$ and $D_{y_2,v}$ runs in time $\mathrm{poly}(d, 1/\eta, (O(1)/\alpha)^{1/\eta}, \log(1/\delta\eta\alpha))$ and returns $v'$ satisfying $\|v' - v\| \le O(\eta\sigma)$ with probability $1 - \delta$.*

*Proof.* Consider a matrix $R$ whose entries $R_{i,j}$ are independently drawn from the uniform distribution over $\pm 1/\sqrt{d}$. and whose diagonals are $1/\sqrt{d}$.

Our goal is to show that with probability at least 99%, the standard deviation of each coordinate of $Rz$ is bounded by $O(\sigma/\sqrt{d})$, i.e., the standard deviation of $Rz \cdot e_i$ is at most $O(\sigma/\sqrt{d})$ for all integer $i$ in $[d]$.

We can then amplify this probability to ensure that the algorithm fails with a probability that is exponentially small.

To see this, observe that $Rz \cdot e_i = r_i \cdot z$, and so $\mathbf{E}_z[r_i \cdot z] = 0$.

$$
\begin{aligned}
\mathbf{E}_z[(r_i \cdot z)^2] &= \sum_{p \in [d], q \in [d]} R_{i,p} R_{i,q} \, \mathbf{E}[z_p z_q] \\
&= \sum_{i=1}^{d} \mathbf{E}[z_i^2]/d + 2 \sum_{p,q \in [d], p < q} R_{i,p} R_{i,q} \, \mathbf{E}[z_p z_q] \\
&\leq (\sigma^2/d) + 2 \sum_{p,q \in [d], p < q} R_{i,p} R_{i,q} \, \mathbf{E}[z_p z_q] \,.
\end{aligned}
$$

We now bound the second term with probability $99\%$ via applying Chebyshev's inequality. Observe that $\mathbf{E}[z_p z_q] \leq \sqrt{\mathbf{E}[z_p^2]\,\mathbf{E}[z_q^2]} \leq \sigma^2$. Since $R_{i,p}$ and $R_{i,q}$ are drawn independently and $p \neq q$, we see that the variables $R_{i,p}R_{i,q}$ and $R_{i,l}R_{i,m}$ pairwise independent for pairs $(p,q) \neq (l,m)$, this implies $\mathbf{Pr}[|\sum_{p,q \in [d], p<q} R_{i,p} R_{i,q} \, \mathbf{E}[z_p z_q]| > T] \leq \frac{O(\sigma^4)}{d\, d^2 T^2}$. By choosing $T = O(\sigma^2/d)$, we see that the right-hand side above is at most $0.001/d$.

A union bound over all the coordinates then tells us that with probability $99\%$, the variance of each coordinate is at most $O(\sigma^2/d)$.

Then, for each coordinate $i$, we can identify $v_i' = v_i \pm O(\eta\sigma/\sqrt{d})$ through an application of Lemma 3.1. Putting these together with probability at least $99\%$, we find $v'$ satisfying $\|v' - v\|^2 \leq O(\eta^2\sigma^2)$.

Changing between these basis representations maintains the quality of our estimate since the new basis contains unit vectors nearly orthogonal to each other. With high probability, the inner products between these are around $O(1/\sqrt{d})$ for every pairwise comparison, so $R$ approximates a random rotation.

*Probability Amplification:* The current guarantee ensures that we obtain a good candidate with a constant probability of success. However, for the final algorithmic guarantee, we need a higher probability of success. To achieve this, we modify the algorithm as follows:

1. Run the algorithm $T$ times, each time returning a candidate $v_i'$ that is, with probability $99\%$, within $O(\eta\sigma)$ distance from the true solution.

2. Construct a list of candidates $C = \{v_1', \ldots, v_T'\}$.

3. Identify a ball of radius $O(\sigma\eta)$ centered at one of the $v_i'$ that contains at least $90\%$ of the remaining points.

4. Return the corresponding $v_i'$ as the final output.

5. If no such $v_i'$ exists, return any vector from $C$.

Let $E$ denote the event that a point is within $O(\eta\sigma)$ to the true solution.

This will succeed with probability $1 - \exp(-T)$. To see why, observe the chance that we recover $(2/3)T$ vectors outside the event $E$ is less than $(0.01)^{2/3\,T}\binom{T}{2/3T} < (0.047)^T\binom{T}{T/2} < (0.047)^T(2^T/\sqrt{T}) < (0.095)^T$. $\qquad\square$

## D  List-Decodable Mean Estimation

This section presents an algorithm for list-decodable mean estimation when the inlier distribution follows $\mathcal{D}_\sigma$. Here, $\mathcal{D}_\sigma$ represents a set of distributions over $\mathbb{R}^d$ defined as $\mathcal{D}_\sigma := \{D \mid \mathbf{E}_D[|x - \mathbf{E}_D[x]|^2] \leq \sigma^2\}$. In our setting, we receive samples from $\xi + z$, where $\mathbf{Pr}[\xi = 0] > \alpha$, where $\alpha$ can be close to 0. Our objective is to estimate the mean with a high degree of precision.

Note that the guarantees provided by prior work do not directly apply to our setting. Prior work examines a more aggressive setting where arbitrary outliers are drawn with a probability of $1 - \alpha$. These outliers might not have the additive structure we have.

Recall the definition of an $(\alpha, \beta, s)$-LDME algorithm:

**Definition D.1** (Algorithm for List-Decodable Mean Estimation). *Algorithm $\mathcal{A}$ is an $(\alpha, \beta, s)$-LDME algorithm for $\mathcal{D}$ (a set of candidate inlier distributions) if with probability $1 - \delta_{\mathcal{A}}$, it returns a list $\mathcal{L}$ of size $s$ such that $\min_{\hat{\mu} \in \mathcal{L}} \|\hat{\mu} - \mathbf{E}_{x \sim D}[x]\| \leq \beta$ for $D \in \mathcal{D}$ when given $m_{\mathcal{A}}$ samples of the kind $z + \xi$ for $z \sim D$ and $\mathbf{Pr}[\xi = 0] \geq \alpha$. If $1 - \alpha$ is a sufficiently small constant less than $1/2$, then $s = 1$.*

We now prove Fact 2.1 which we restate below for convenience.

**Fact D.2** (List-decoding algorithm). *There is an $(\alpha, \eta\sigma, \tilde{O}((1/\alpha)^{2/\eta^2}))$-LDME algorithm for the inlier distribution belonging to $\mathcal{D}_\sigma$ which runs in time $\tilde{O}(d(1/\alpha)^{2/\eta^2})$ and succeeds with probability $1 - \delta$. Conversely, any algorithm which returns a list, one of which makes an error of at most $O(\eta\sigma)$ in $\ell_2$ norm to the true mean, must have a list whose size grows exponentially in $1/\eta$.*

*If $1 - \alpha$ is a sufficiently small constant less than half, then the list size is 1 to get an error of $O(\sqrt{1 - \alpha}\, \sigma)$.*

*Proof.* **Algorithm:** Consider the following algorithm:

1. If $\alpha < c$ and $\eta > \sqrt{\alpha}$: Run any stability-based robust mean estimation algorithm from Diakonikolas et al. [2020b] and return a singleton list containing the output of the algorithm.

2. Otherwise, for integer each $i \in [1, 100(1/\alpha)^{2/\eta^2} \log(1/\delta)^2]$ sample $1/\eta^2$ samples and let their mean be $\mu_i$.

3. Return the list $\mathcal{L} = \{\mu_i \mid i \in [1, 100(1/\alpha)^{2/\eta^2} \log(1/\delta)^2]\}$.

If the algorithm returns in the first step, then the guarantees follow from the guarantees of the algorithm for robust mean estimation from Diakonikolas et al. [2020b] (Proposition 1.5 on page 4).

Otherwise, observe that the probability that every one of $1/\eta^2$ samples drawn is an inlier, is $\alpha^{1/\eta^2}$.

Hence, with probability $1 - \delta$ we see that if we draw $1/\eta^2$ samples $O((1/\alpha)^{2/\eta^2} \log(1/\delta)^2)$ times, there are at least $O(\log(1/\delta))$ sets of samples containing only inliers. Then, the mean of one of these concentrates to an error of $O(\eta\sigma)$ by an application of Lemma A.2. More precisely, Lemma A.2 ensures that with probability 99%, the mean of a set of $1/\eta^2$ inliers concentrates up to an error of $O(\eta\sigma)$. Repeating this $\log(1/\delta)$ times, we get our result.

**Hardness:** To see that the list size must be at least $\exp(1/\eta)$, consider the set of inlier distributions given by $\{D_s \mid s \in \{\pm 1\}^d\}$ where each $D_s$ is defined as follows: $D_s$ is a distribution over $\mathbb{R}^d$ such that each coordinate independently takes the value $s_i$ with probability $1/d$, and 0 otherwise.

Each $D_s$ defined above belongs to $\mathcal{D}_{\sqrt{1-1/d}}$ since $\mathbf{E}_{x \sim D_s}[x] = s/d$ and

$$\sigma^2 := \mathop{\mathbf{E}}_{x \sim D_s}[\|x - s/d\|^2] = \sum_{i=1}^d \mathop{\mathbf{E}}_{x_i \sim (D_s)_i}[(x_i - s_i/d)^2]$$

$$= \sum_{i=1}^d (1 - 1/d)(1/d)^2 + (1/d)(1 - 1/d)^2 = (1 - 1/d).$$

We will set the oblivious noise distribution for each $D_s$ to be $-D_s$. Our objective is to demonstrate that the distribution of $D_s - D_s$ is the same for all $s$ and is independent of $s$. This means that we cannot identify $s$ by seeing samples from $D_s - D_s$.

Then, since the means of $D_s$ and $D'_s$ for any distinct pair $s, s' \in \{\pm 1\}^d$ differ by at least $1/d$, if we set $d = 1/\eta$ we see that there are $2^{1/\eta}$ possible different values of the original mean, each pair being at least $\eta$ far apart, which is larger than $\eta\sigma^2 = \eta(1 - \eta)$.

We can assume, without loss of generality, that $s = \mathbf{1}$, where $\mathbf{1}$ represents the all-ones vector. Each coordinate of $D_s$ can be viewed as a random coin flip, taking the value 0 with probability $1 - 1/d$ and 1 with probability $1/d$.

The probability of obtaining the all-zeros vector is given by $(1 - 1/d)^d$, which approaches a constant value for sufficiently large $d$, and so, $\mathbf{Pr}_{x \sim D_s}[x = 0] \geq 0.001$, i.e., the $\alpha$ for the oblivious noise is at least a constant. In fact, it can be as large as $1/e > 0.35$ for large enough $d$.

Let the oblivious noise be $-D$. Now, consider the distribution of $x + y$, where $x$ follows the distribution $D$ and $y$ follows the distribution $-D$. If we focus on the first coordinate, $x_1 + y_1$, we observe that it follows a symmetric distribution over $\{-1, 0, 1\}$ which does not depend on $s_1$. Also, each coordinate exhibits the same distribution, and they are drawn independently of one another. Hence, the final distribution is independent of $s$, so we get our result. $\quad\square$

# E  Proof of Corollary 4.2

Below, we restate Corollary 4.2 for convenience.

**Corollary E.1.** *Given access to oblivious noise oracle $\mathcal{O}_{\alpha, \sigma, f}$, a $(O(\eta\sigma), \epsilon)$-inexact-learner $\mathcal{A}_G$ running in time $T_G$, there exists an algorithm that takes $\mathrm{poly}((1/\alpha)^{1/\eta^2}, (O(1)/\eta)^{1/\eta}, \log(T_G/\delta\eta\alpha))$ samples, runs in time $T_G \cdot \mathrm{poly}(d, (1/\alpha)^{1/\eta^2}, (O(1)/\eta)^{1/\eta}, \log(1/\eta\alpha\delta))$, and with probability $1 - \delta$ returns a list $\mathcal{L}$ of size $\tilde{O}((1/\alpha)^{1/\eta^2})$ such that $\min_{x \in \mathcal{L}} \|\nabla f(x)\| \leq O(\eta\sigma) + \epsilon$. Additionally, the exponential dependence on $1/\eta$ in the list size is necessary.*

*Proof.* This follows by substituting the guarantees of Fact 2.1 for the algorithm $\mathcal{A}_{ME}$ in Theorem 1.4. $\quad\square$

# F  Varying Noise Assumptions

In this section, we demonstrate that changing some distributional assumptions of the oblivious noise $\xi$ can yield better guarantees or simpler algorithms.

## F.1  Stronger Concentration on $\xi$ Yields Polynomial Dependence on $1/\eta$

We show that if the oblivious noise has bounded tails we can avoid the exponential dependence on $1/\eta$, which we have shown in Appendix D to be necessary under our standard assumptions.

For the sake of demonstration, we will assume the standard assumptions of heavy-tailed stochastic optimization where the mean of $\xi$ is equal to zero and the variance of $\xi$ is upper bounded by $\sigma$, as is done in Gorbunov et al. [2020]. In this specific setting, our algorithm remains the same but our analysis becomes tighter in Item 2 of Claim C.4. Our original analysis yields an exponential dependence on $1/\eta$ due to the upper bound on $k$, yet with concentration on the noise, we can show that $k = \mathrm{poly}(1/\alpha, 1/\eta)$.

Due to our specific construction of $\tilde{P}(k)$ in the proof of Claim C.4 of Item 1, we have that $k\tilde{P}(k) \leq O(k^{-1/3})$ and $\alpha/2 \leq \tilde{P}(1) \leq \sum_{j=1}^k \tilde{P}(j)$. Thus, when $k \geq O(1/(\alpha\eta)^3)$, the inequality of Item 2 is satisfied. The fact that $\alpha/2 \leq \tilde{P}(1)$ is given, and thus, what remains is to show $\tilde{P}(k) \leq k^{-4/3}$, which follows by splitting the summation in $\tilde{P}(i, z)$ over $j = 1, ..., k-1$ into $j \in [1, k^{2/3}]$ and $j \in (k^{2/3}, k-1]$. The sum over $[1, k^{2/3}]$ is upper bounded by $O(k^{-4/3})$ and the sum over $(k^{2/3}, k-1]$ is upper bounded by $O(A^{-2}k^{-4/3}) = O(\alpha k^{-4/3})$ using Chebyshev's inequality. Therefore, $\tilde{P}(k) \leq k^{-4/3}$ and $k = \mathrm{poly}(1/\alpha, 1/\eta)$ in Claim 3.3.

Ultimately, it would be an interesting direction to identify conditions necessary and sufficient for a polynomial dependence on $1/\eta$.

## F.2  Small Fraction of Adversarial Corruptions

If, on the other hand, at each point you have access to gradients $v(x)$ such that with probability $1 - \epsilon$ you see $\nabla f(x, \gamma)$, which concentrates around $\nabla f(x, \gamma)$ and with probability $\epsilon$ you see an arbitrary point, then one can use robust mean estimation instead of our location estimation algorithm and recover guarentees which are correct upto an error of $O(\sqrt{\epsilon})$.

In this case, there is no need for list-decoding or robust location estimation.

