Since $\Pr[x = 0] \geq \alpha$, we see that $\Pr[x < 0] \leq 1/2 - \alpha$,

and so $\Pr[x + y < -O(\sigma/\sqrt{\alpha})] < (1 + \alpha)(0.5 - \alpha) = 0.5 - \alpha + 0.5\alpha - \alpha^2 = 0.5 - 0.5\alpha - \alpha^2 < 0.5$.

The upper bound follows similarly.

Since $\Pr[x = 0] \geq \alpha$ and $\Pr[|y| < O(\sigma/\sqrt{\alpha})] \geq 1 - \alpha$, we see $\Pr[|x + y| < O(\sigma/\sqrt{\alpha})] \geq \alpha/2$. Hoeffding's inequality (Lemma A.1) now implies that the empirical median also satisfies the above upper bound as long as the number of samples is greater than $O(1)/\alpha^2 \log(1/\delta)$.

To see that this is tight, consider the distribution centered at 0, whose density function is $2/(y + 2)^3$ in the range $[1, \infty)$, and is 0 otherwise.

Call this $D_{y^{-3}}$. Observe that $\Pr_{D_{y^{-3}}}[z > t] < O(1) \int_t^\infty y^{-3} dy = C/t^2$.

Let $x$ be a symmetric distribution whose distribution takes the value 0 with probability $\alpha$ and takes the values $\pm \alpha^{-1/2}100C^{1/2} + 10$ with probability $0.5(1 - \alpha)$.

We show that the median of the distribution of $x + y$ where $y$ is drawn from $D_{y^{-3}}$, is larger than $\Omega(\alpha^{-1/2})$.

To see this, we show that the probability that $x + y$ takes a value smaller than $100\alpha^{-1/2}C^{1/2}$ is less than half, implying that the median has to be larger than this quantity.

$x$ takes three values. Note that $y + 100\alpha^{-1/2}C^{1/2} + 12$ places no mass in the region $(-\infty, 100\alpha^{-1/2}C^{1/2} + 10]$. So to estimate the probability that $x + y$ takes a value smaller than $100\alpha^{-1/2}C^{1/2} + 10$, we only need to consider contributions from the other two possible values. By

choosing $\alpha$ small enough, so that $100\alpha^{-1/2}C^{1/2} > 10$, we see

$$
\begin{aligned}
&\Pr[x + y < 100\alpha^{-1/2}C^{1/2} + 10] \\
&< 0.5(1-\alpha)\Pr[y < 200\alpha^{-1/2}C^{1/2} + 20] + \alpha\Pr[y < 100\alpha^{-1/2}C^{1/2} + 10] \\
&< 0.5(1-\alpha)(1 - C/(200\alpha^{-1/2}C^{1/2} + 20)^2) + \alpha(1 - C/(100\alpha^{-1/2}C^{1/2} + 10)^2) \\
&< 0.5(1-\alpha)(1 - \alpha^{1/2}/(400)^2) + \alpha \\
&< 0.5 + 0.5\alpha - \alpha^{1/2}/8 \cdot (400^2) \, .
\end{aligned}
$$

We are done when $0.5\alpha - \alpha^{1/2}/8 \cdot (400^2) < 0$, this happens for $\alpha^{1/2} < 2/(8 \cdot (400^2))$. $\qquad\square$

## B.2   Useful Lemma for Finer Estimation

In the second step of our location-estimation lemma, we refine the estimate of $t$. To do this, we first re-center the distributions based on our rough estimate, so that the shift after re-centering is bounded. Then, we identify an interval $I$ centered around 0 such that, when conditioning on $\xi + z$ falling within this interval, the expected value of $\xi + z$ remains the same as when conditioning on $\xi$ falling within the same interval. This expectation will help us get an improved estimate, which we use to get an improved re-centering of our original distributions, and repeat the process.

To identify such an interval, we search for a pair of bounded-length intervals equidistant from the origin (for e.g. $(-10\sigma, -5\sigma)$ and $(5\sigma, 10\sigma)$) that contain very little probability mass. By doing so, when $z$ is added to $\xi$, the amount of probability mass shifted into the interval $(-5\sigma, 5\sigma)$ $z$ remains small.

In this subsection, we prove Lemma B.2, which states that any positive sequence which has a finite sum must eventually have one small element. The lemma also gives a concrete upper bound on which element of the sequence satisfies this property.

**Lemma B.2.** $a_i \geq 0$ for all $i$ and $\sum_{i=1}^{\infty} a_i < C$ for some constant $C$. Also, suppose we have $\eta \in [0, 1]$. Then there is an $i$ such that $1 \leq L < i < (C/a_0 + L)^{1/\eta}$ such that $ia_i < \eta\sum_{j=1}^{i} a_j$.

Consider a partition of the reals into length $L$ intervals. In our proof, we will use Lemma B.2 on the sequence $a_i$, where $a_i$ corresponds to an upper bound on the mass of $\xi$ contained in the $i$-th intervals equidistant from the origin on either side, and the mass that crosses them (i.e., the mass of $\xi$ that is moved either inside or out of the interval when $z$ is added to it).

We need the following calculation to prove Lemma B.2.

Notation: For integer $i \geq 1$ and $\eta \in (0, 1)$, define $(i - \eta)! := \Pi_{j=1}^{i}(j - \eta)$.

**Fact B.3.** Let $A_k := 1 + \sum_{t=1}^{k-1} \frac{\eta(t-1)!}{(t-\eta)!}$. Then, for $k \geq 2$, $A_k = (k-1)!/(k-1-\eta)!$.

*Proof.* We prove this by induction. By definition, our hypothesis holds for $A_2$ because $A_2 = 1 + \eta/(1-\eta) = 1/(1-\eta) = (2-1)!/(2-1-\eta)!$. Suppose it holds for all $2 \leq t \leq k$. We then show that it holds for $t = k + 1$.

$$
\begin{aligned}
A_{k+1} &= 1 + \sum_{t=1}^{k} \frac{\eta(t-1)!}{(t-\eta)!} = A_k + \frac{\eta(k-1)!}{(k-\eta)!} \\
&= \frac{(k-1)!}{(k-1-\eta)!} + \frac{\eta(k-1)!}{(k-\eta)!} = \frac{(k-1)!}{(k-1-\eta)!}\left(1 + \frac{\eta}{k-\eta}\right) \\
&= \frac{(k-1)!}{(k-1-\eta)!}\frac{k}{k-\eta} = \frac{k!}{(k-\eta)!}.
\end{aligned}
$$

$\qquad\square$

*Proof of Lemma B.2* Let $U = (C/a_0 + L)^{1/\eta}$ and suppose towards a contradiction that there is no such $i$ that satisfies the lemma. Specifically, all integers $i \in [1, U]$, we will assume that for $ia_i \geq \eta\sum_{j=1}^{i} a_j$. We then show that this implies $i^{1-\eta}a_i \geq \eta a_0$ for all $i$ in the range.

Consider the inductive hypothesis on $t$ given by $a_t \geq \eta \frac{(t-1)!}{(t-\eta)!} \cdot a_0$. The base case when $t = 1$ is true since $a_1 \geq \eta a_0 / (1 - \eta)$ by our assumption. Suppose the inductive hypothesis holds for integers $t \in [1, k-1]$. We show this for $t = k$ below.

$$a_k \geq \frac{\eta}{k-\eta} \sum_{t=0}^{k-1} a_t$$

$$\geq \frac{a_0 \eta}{k-\eta} \left( 1 + \sum_{t=1}^{k-1} \frac{\eta(t-1)!}{(t-\eta)!} \right)$$

$$= a_0 \eta \frac{(k-1)!}{(k-\eta)!} \ .$$

The final equality follows from Fact B.3 which states that $(k-1)!/(k-1-\eta)! = 1 + \sum_{t=1}^{k-1} \frac{\eta(t-1)!}{(t-\eta)!}$. Simplifying this further, we see that since $(i - \eta) \geq i \exp(-\eta/i)$ for all $i \in [1, k]$,

$$a_k \geq a_0 \eta \frac{(k-1)!}{(k-\eta)!}$$

$$\geq a_0 \eta \frac{(k-1)!}{k! \exp(-\eta/k)}$$

$$\geq a_0 \eta \, (1/k) \, (1/\exp(-\eta(\sum_{i=1}^{k} 1/i)))$$

$$\geq a_0 \eta \, (1/k) \, (1/\exp(-\eta \log(k)/20))$$

$$\geq (a_0/2)\eta \, (1/k^{1-\eta/20}) \ .$$

Finally, observe that

$$C = \sum_{i=L}^{U} a_i > a_0 \eta \sum_{i=L}^{U} (1/i^{1-\eta/20})$$

$$> a_0 \eta \int_{L}^{U} (1/x^{1-\eta}) \, dx$$

$$= a_0(U^\eta - L^\eta).$$

If $a_0(U^\eta - L^\eta) > C$, we have a contradiction, since $\sum_{i=L}^{\infty} a_i < C$. This follows when $U > (C/a_0 + L)^{1/\eta}$. $\

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

 less than $O(\eta/i)$ by using $\text{poly}((O(1)/\eta\alpha)^{1/\eta}, \log(1/\delta\alpha\eta))$ samples from $\xi + z$ and*
649 *$\xi + z'$.*

650 We defer the proof of Claim C.4 to Appendix C.2, and continue with our proof showing that
651 $\mathbf{E}[\xi + z \mid |\xi + z| \leq A\sigma k] \approx \mathbf{E}[\xi \mid |\xi| \leq A\sigma k]$ for $k$ satisfying the conclusions of Claim C.4. To this
652 end, observe the following for $f(\xi, z)$ being either 1 or $\xi + z$.

$$|\mathbf{E}[f(\xi, z) \, \mathbf{1}(|\xi| \leq \sigma i)] - \mathbf{E}[f(\xi, z) \, \mathbf{1}(|\xi + z| \leq \sigma i)]|$$
$$\leq |\mathbf{E}[f(\xi, z) \, \mathbf{1}(|\xi + z| > \sigma i) \, \mathbf{1}(|\xi| \leq \sigma i)]| + |\mathbf{E}[f(\xi, z) \, \mathbf{1}(|\xi + z| \leq \sigma i) \, \mathbf{1}(|\xi| > \sigma i)]|.$$

653 By setting $f(\xi, z) := 1$ and considering the case where $i = k$ satisfies the conclusions of Claim C.4,
654 we can bound the "error terms"

655 $\Pr[|\xi + z| \leq A\sigma k$ and $|\xi| > A\sigma k]$ and $\Pr[|\xi + z| > A\sigma k$ and $|\xi| \leq A\sigma k]$

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

$$\Pr[|\xi + z| < A\sigma(k+5)] + \Pr[|\xi + z'| < A\sigma(k+5)]$$

$$= \Pr[|\xi + z| < A\sigma(k-4)] + \Pr[|\xi + z'| < A\sigma(k-4)]$$

$$+ \Pr[|\xi + z| \in A\sigma(k-4, k+5)] + \Pr[|\xi + z'| \in A\sigma(k-4, k+5)]$$

$$\le \Pr[|\xi + z| < A\sigma(k-4)] + \Pr[|\xi + z'| < A\sigma(k-4)]$$

$$+ \tilde{P}(k)$$

$$\le \Pr[|\xi + z| < A\sigma(k-4)] + \Pr[|\xi + z'| < A\sigma(k-4)]$$

$$+ O(\eta/k)(\Pr[|\xi + z| < A\sigma(k+5)] + \Pr[|\xi + z'| < A\sigma(k+5)]) \,.$$

737 Rearranging the inequality and by scaling $\eta$ such that that $O(\eta/k) \le 1/2$, we see that $\Pr[|\xi + z| <$
738 $A\sigma(k+5)] + \Pr[|\xi + z'| < A\sigma(k+5)] = O(\Pr[|\xi + z| < A\sigma(k-4)] + \Pr[|\xi + z'| < A\sigma(k-4)]) =$
739 $O(\Pr[|\xi + z| < A\sigma k] + \Pr[|\xi + z'| < A\sigma k])$, completing our proof of Item 3.

740 *Proof of Item 4:*

Finally, to see Item 4 holds, observe that $0 < \tilde{P}(i) < O(1)$. Let $B = (O(1)/\eta\alpha)^{1/\eta}$ denote the maximum index before which we can find a $k$ such that $k\tilde{P}(k) \leq \eta \sum_{i=1}^{k} \tilde{P}(i)$. Now, To estimate $\tilde{P}(i)$ empirically, we partition the interval $(-BA\sigma, BA\sigma)$ into $B$ intervals of length $A\sigma$ each, and estimate the probability of $\xi + z$ falling in each interval. If we estimate each of these probabilities to an accuracy of $\eta/(100\,B)$, we can estimate $\tilde{P}(i)$ to an accuracy of $O(\eta/i)$.

An application of Hoeffding's inequality (Lemma A.1) tells us that each estimate will require $O(B^2/\eta^2 \log(1/\delta))$ samples. Taking a union bound over all these intervals, we see that we will require $O(B^2/\eta^2 \log(B/\delta))$ samples.

Finally, another union bound over each $i \in [0, B]$ implies that we will need $O(B^2/\eta^2 \log(B^2/\delta))$ samples. Substituting the value of $B$ back in, we see that this amounts to requiring $(O(1)/\eta\alpha)^{2/\eta} \log(1/\eta\alpha\delta)$ samples.

Estimating $\tilde{P}(i)$ will take time polynomial in the number of samples, and so we take time $\tilde{O}(O(1)/\eta\alpha)^{O(1)/\eta}$. $\qquad\square$

## C.3 High-dimensional Noisy Location Estimation

In this section, we explain how to use our one-dimensional location estimation algorithm to get an algorithm for noisy location estimation in $d$ dimensions.

The algorithm performs one-dimensional location estimation coordinate-wise, after a random rotation.

We need to perform such a rotation to ensure that every coordinate has a known variance bound of $\sigma/\sqrt{d}$.

---

**Algorithm 5** High-dimensional Location Estimation: ShiftHighD$(S_1, S_2, \eta, \sigma, \alpha)$

---

**input:** Sample sets $S_1, S_2 \subset \mathbb{R}^d$ of size $m$, $\eta \in (0,1)$, $\sigma > 0$, $\alpha$

1. Sample $R_{i,j}$ i.i.d. from the uniform distribution over $\{\pm 1/\sqrt{d}\}$ for $i, j \in [d]$
2. Represent $S_1$ and $S_2$ in the basis given by the rows of $R$: $r_1, \ldots, r_d$.
3. **for** $i \in [d]$ **do**
   $\quad \mid \quad v_i' := \text{Shift1D}(S_1 \cdot e_i, S_2 \cdot e_i, \eta, O(\sigma/\sqrt{d}), \alpha)$
**end**
4. Change the representation of $v'$ back to the standard basis.
5. *Probability Amplification:* Repeat steps 1-4, $T := O(\log(1/\delta))$ times to get $C := \{v_1', \ldots, v_T'\}$
6. Find a ball of radius $O(\eta\sigma)$ centered at one of the $v_i'$ containing $> 90\%$ of $C$. If such a vector exists, set $v'$ to be this vector. Otherwise set $v'$ to be an arbitrary element of $C$.
5. Return $v'$.

---

**Lemma C.5** (Location Estimation). *Let $y_i := \xi + z_i$ for $i \in \{1, 2\}$ where $\Pr[\xi = 0] \geq \alpha$ and $z_i \sim D_i$ are distributions over $\mathbb{R}^d$ satisfying $\mathbf{E}_{D_i}[x] = 0$ and $\mathbf{E}_{D_i}[\|x\|^2] \leq \sigma^2$. Let $v \in \mathbb{R}^d$ be an unknown shift. There is an algorithm (Algorithm 5), which draws $m = \text{poly}((O(1)/\eta\alpha)^{1/\eta}, \log(1/\delta\epsilon\alpha))$ samples each from $y_1$ and $