# OpenReview forum: "First Order Stochastic Optimization with Oblivious Noise"
_NeurIPS.cc/2023/Conference — NeurIPS 2023 poster_

### Official Review · Reviewer_2f9g · 2023-07-04

**Soundness:** 4 excellent
**Presentation:** 4 excellent
**Contribution:** 4 excellent
**Rating:** 7
**Confidence:** 3

**Summary:**

This paper initiates the study of stochastic optimization with oblivious noise that might be biased and have unbounded variance, which is a generalization of the heavy-tailed noise setup. The key assumption regarding the oblivious noise is that it assumes a value of $0$ with a probability within the range of $0 \leq \alpha \leq 1$, which can be interpreted as the fraction of inliers. Notably, when $\alpha \leq 1/2$, it is proven to be information-theoretically impossible to find an approximate stationary point of the function. To address this challenge, the authors incorporate the concept of list-decoding into the framework of stochastic optimization, and focus instead on identifying a list of points where at least one of them is an approximate stationary point.

Technically, this paper presents an equivalence between list-decodable stochastic optimization with oblivious noise and list-decodable mean estimation problem leveraging a technique known as noisy location estimation. The analysis of list-decodable stochastic optimization with oblivious noise is conducted by examining the list-decodable mean estimation problem.

**Strengths:**

This paper investigates an important setting of stochastic optimization introduces a fresh perspective by introducing the concept of list-decodable stochastic optimization. The definition of this new framework is not only intuitive and well-motivated by practical problems but also exhibits elegance from a theoretical standpoint, given how weak the assumptions on the noise model are. The algorithms presented in the paper, along with their corresponding proofs, are intricate and highly nontrivial from a technical standpoint. Nevertheless, the authors have succeeded in presenting the analyses in a well-organized manner, ensuring that they are generally not hard to follow.

**Weaknesses:**

1. As pointed out in the paper, an exponential dependence on $1/\eta$ is necessary in the list-size, which can mildly impact the overall appeal of the results. This dependency may introduce some considerations regarding scalability and practicality.

2. The framework presented in this paper is inherently abstract, and there is a lack of clarity concerning the algorithm's performance in concrete examples, including scenarios with more specific theoretical settings that incorporate additional assumptions, as well as practical problem domains.

**Questions:**

Correspondingly, I have the following questions that could possibly make the results even stronger if addressed:
1. Can the exponential dependence on $\eta$ be mitigated by making slight adjustments to the original definition of list-decodable stochastic optimization? For instance, are there additional assumptions that can be incorporated or specific parameter regimes that can be adjusted to reduce this exponential dependency?

2. Are there more concrete applications of list-decodable stochastic optimization methods?

3. Minor comment: I saw the term "convex" in the caption of Algorithm 2. I assume this is a typo and convexity is not needed in the proof, right?

**Limitations:**

Not relevant in my opinion.

---

> ### Author Rebuttal · Authors · 2023-08-10
>
> We first give a common response, and then addresses the specific concerns of the reviewer.
>
> **Common Response**
>
> To summarize, our paper investigates a generalization of the heavy-tailed stochastic optimization problem. As highlighted by reviewers M73n, NDsr, and 2f9g, our work addresses a challenging and well-motivated scenario with minimal noise assumptions. It introduces a novel perspective on the connection between stochastic optimization and problems such as list-decodable mean estimation and location estimation, employing an innovative efficient reduction (reviewer 2f9g and FMyo). A significant contribution of our work is the resolution of the noisy location estimation problem. The proof of our algorithm’s guarantee is novel and technically involved (reviewers 2f9g and NDsr). Notably, our method produces a single candidate for a sufficiently small constant fraction of outliers, allowing for a direct comparison with previous approaches in stochastic convex optimization.
>
> We will fix the typos and address minor comments mentioned by the reviewers in the final version of the manuscript. We now address more specific concerns of the reviewer.
>
> **Response to Reviewer 2f9g**
>
> Thank you for your insightful and detailed review. We address your questions below.
>
> 1. **Exponential dependence on $1/\eta$ and concrete runtime in specific settings:**
> Indeed, this is an interesting question, and we can mitigate the exponential dependence on $1/\eta$ if we make additional assumptions on the oblivious noise $\xi$.
> For the sake of demonstration, we will assume the standard assumptions of heavy-tailed stochastic optimization where the mean of $\xi$ is equal to zero and the variance of $\xi$ is upper bounded by $\sigma$, as is done in [1]. In this specific setting, our algorithm remains the same but our analysis becomes tighter in Claim 3.3 (or Claim C.4) item 2. Our original analysis yields an exponential dependence on $1/eta$ due to the upper bound on $k$, yet with concentration on the noise, we can show that $k = poly(1/\alpha, 1/\eta)$.
> Due to our specific construction of $\tilde{P}(k)$ in L710, we have that $k\tilde{P}(k) \le O(k^{-1/3})$ and $\alpha/2 \le \tilde{P}(1) \le \sum_{j=1}^k \tilde{P}(j)$. Thus, when $k \ge O(1/(\alpha \eta)^3)$, the inequality of item 2 is satisfied. The fact that $\alpha/2 \le \tilde{P}(1)$ is given in L719 so what remains is to show $\tilde{P}(k) \le k^{-4/3}$, which follows by splitting the summation in $\tilde{P}(i,z)$ over $j=1…k-1$ (defined in L709) into $j \in [1, k^{2/3}]$ and $j \in (k^{2/3}, k-1]$. The sum over $[1, k^{2/3}]$ is upper bounded by $O(k^{-4/3})$ and the sum over $(k^{2/3}, k-1]$ is upper bounded by $O(A^{-2}k^{-4/3}) = O(\alpha k^{-4/3})$ using Chebyshev’s inequality. Therefore, $\tilde{P}(k) \le k^{-4/3}$ and $k = poly(1/\alpha, 1/\eta)$ in Claim 3.3.
> Ultimately, it would be an interesting direction to identify conditions necessary and sufficient for a polynomial dependence on $1/\eta$.
>
> 2. **More concrete application of list-decodable stochastic optimization:**
> We think of list-decoding as a natural consequence of having more than half the points being outliers. As mentioned in [2], there are practical settings where it is possible to get a very small set of verified samples which could be used to determine which of the multiple answers is the correct one (the "semi-verified" setting).
>
> 3. **“Convex” in the caption of Algorithm 2:**
> Yes, this is a typo.
>
> [1] E. Gorbunov, M. Danilova, and A. Gasnikov. Stochastic optimization with heavy-tailed noise via accelerated gradient clipping. Advances in Neural Information Processing Systems, 33:15042–15053, 2020.
>
> [2] M. Charikar, J. Steinhardt, and G. Valiant. Learning from untrusted data. Proceedings of the 49th Annual ACM SIGACT Symposium on Theory of Computing. 2017.

---

> > ### Comment · Reviewer_2f9g · 2023-08-14
> >
> > I would like to thank the authors for the detailed rebuttal. I remain my rating.

---

### Official Review · Reviewer_5dvc · 2023-07-06

**Soundness:** 2 fair
**Presentation:** 1 poor
**Contribution:** 2 fair
**Rating:** 4
**Confidence:** 3

**Summary:**

The paper presents an algorithm for first-order stochastic optimization where the algorithm has access to an oracle that returns a noisy version of the gradient of the objective function. The considered noise model includes two components: A bounded-variance observation noise (which is the typical well-studied type of noise), and oblivious outliers noise $\xi$ satisfying $Pr[\xi = 0] >= \alpha$. Furthermore, the distribution of the oblivious noise \xi does not need to be symmetric.

It is shown that if the fraction of inliers is below 1/2, it is information-theoretically impossible to give a unique solution. This is why the authors consider a list-decodable learner where the learner returns a list of solutions, one of which is guaranteed to be good. The authors show that if the fraction of inliers is sufficiently close to 1, then the algorithm can recover a single solution.

**Strengths:**

Designing learning algorithms which are robust against adversarial or semi-adversarial type of noise is very important. The setup that is considered in this paper is original (as far as I can tell).

**Weaknesses:**

I found the paper to be generally not very well written. The notation is a bit confusing in several places (e.g., check the question regarding line 203 below), and the writing style can be sometimes too informal.

One thing that I found crucially missing is the clear and formal statement of the problem and the clear statements of the assumptions. For example, what are the properties of the function $f(\gamma,x)$? The only property that I found is that $f(x) = E_{\gamma}[f(\gamma,x)]$ must be $L$-smooth. However, this is clearly not enough to even guarantee the existence of a stationary point. For example, consider $x\in \mathbb{R}$ (i.e., one dimension) and define $f(\gamma,x) = x$. In this case, we have $f(x) = x$ and hence $\nabla f(x) = 1$ fo all $x$ and there is no stationary point.

Typos:
- Page 4, line 175: "we can a generate list" -> "we can generate a list"

**Questions:**

- What are the properties of the function $f(\gamma,x)$ which are needed for the main result to hold?
- Page 4, line 203: Is $\xi$ in $\xi + y' + t$ the same as the $\xi$ in $\xi + y$, or is it an independent instance? It seems from the following discussion that the authors consider an independent instance. If this is the case, please write $\xi' + y' + t$.
- Page 7, line 309: What is $L$? Is it the same as the $L$ of Section 2? But in Section 3 we don't have a parameter $L$ for location estimation.
- There doesn't seem to be a proof for Claim 3.3 (even in the appendices).

**Limitations:**

No concerns regarding potential societal impact of this work.

---

> ### Author Rebuttal · Authors · 2023-08-10
>
> We first give a common response, and then addresses the specific concerns of the reviewer.
>
> **Common Response**
>
> To summarize, our paper investigates a generalization of the heavy-tailed stochastic optimization problem. As highlighted by reviewers M73n, NDsr, and 2f9g, our work addresses a challenging and well-motivated scenario with minimal noise assumptions. It introduces a novel perspective on the connection between stochastic optimization and problems such as list-decodable mean estimation and location estimation, employing an innovative efficient reduction (reviewer 2f9g and FMyo). A significant contribution of our work is the resolution of the noisy location estimation problem. The proof of our algorithm’s guarantee is novel and technically involved (reviewers 2f9g and NDsr). Notably, our method produces a single candidate for a sufficiently small constant fraction of outliers, allowing for a direct comparison with previous approaches in stochastic convex optimization.
>
> We will fix the typos and address minor comments mentioned by the reviewers in the final version of the manuscript. We now address more specific concerns of the reviewer.
>
> **Response to Reviewer 5dvc**
>
> Thank you for taking the time to review our paper and acknowledging our original setup. The main body of the paper attempts to communicate the ideas involved and hence sometimes is somewhat informal.
>
> Here are answers to your more specific questions:
>
> 1. **Properties of $f(\gamma, x)$**:
> Thank you for pointing this out. Our main contribution is a general method for de-biasing gradients (robust gradient estimator) which can be used along-side any standard stochastic optimization method that requires unbiased gradients. Our de-biasing method does not require any assumptions beyond those mentioned in Definition 1.1 (and the fact that the noise is independent). However, there may be additional assumptions that the black-box optimization algorithm requires.
> As stated on L282-L284 of the main body, if we use the algorithm from [1] for our underlying learner, in addition to the function being $L$-smooth, we would also need that a global minimum exists and that the initial function value has a bounded gap to reach a point with small average gradient. We will mention this assumption in the definition.
>
>
> 2. **$\xi + y’ + t$ and $\xi + y$**:
> We think of $\xi, y, y’$ as random variables. Indeed, it is an independent instance.
>
>
> 3. **$L$ on Line 309**:
> It is not the same $L$ as in Section 2. One may think of $L$ as a free parameter here, this holds for any $L > 0$. It comes from Lemma B.2 in the supplementary which is a lemma used in Claim C.4 (more formal version of Claim 3.3). As can be seen in Claim C.4, $L = 1/\eta \alpha$ suffices for our case.
>
>
> 4. **Proof of Claim 3.3**:
> This is claim C.4 in the appendix, the proof of which is in section C.2 (starting at L690 in the supplementary material).
>
> [1] A. Ajalloeian and S. U. Stich. Analysis of sgd with biased gradient estimators. arXiv preprint arXiv:2008.00051, 2020.

---

> > ### Comment · Reviewer_5dvc · 2023-08-18
> >
> > I would like to thank the authors for their reply.
> >
> > Regarding my point about $L$ in line 309: If this is a free parameter, you can just say something like "Let $L>0$ be such that ...". If you do not mention anything the reader will assume that this is a predefined parameter and will keep wondering what it is. Furthermore, since this is not the same $L$ as in Section 2, please use another letter.
> >
> > I would like to mention that when I reviewed the paper, I did not carefully read the supplementary material and only read the main paper. I invite the chairs to take this into consideration and perhaps give less weight to my review. I was a bit discouraged from carefully reading the supplementary material because I found the main text to be non-very-well-written and a bit informal.

---

### Official Review · Reviewer_M73n · 2023-07-06

**Soundness:** 3 good
**Presentation:** 3 good
**Contribution:** 3 good
**Rating:** 7
**Confidence:** 4

**Summary:**

The paper considers the problem of stochastic optimization with oblivious noise. Here, one receives noisy gradients of a non-convex function $f(x) = \mathbb{E} [f(x, \gamma)]$ (where $\gamma$ is bounded variance observational noise) and the noisy gradient samples are generated as follows $\nabla_x f(x, \gamma) + \xi$ where $\xi$ is generated independently of $\gamma$ and $x$ with the only restriction that $\mathbb{P} (\xi = 0) \geq \alpha$. The paper specifically focuses on the setting where $\alpha \ll 1 / 2$. Under such mild restrictions on $\xi$, it is impossible to recover a single point which is guaranteed to be near-stationary. However, in line with recent results on list-decodable robust estimation, the paper shows that one can recover a list of estimates one of which is approximately stationary.

Technically, the paper builds on two recent results on robust estimation. The first is SEVER which is a robust stochastic estimation algorithm focusing on the setting when $\alpha \to 1$. In this setting, it is possible to leverage recent robust estimation algorithms to clean the observed gradients and recover a good approximation to the true gradient (up to the degree determined by $1 - \alpha$) and then utilize this approximate gradient to find a stationary point. The second is a recent line of work on list-decodable mean estimation. Here, one receives corrupted samples from a high-dimensional distribution where $\alpha$ fraction of points are from the true distribution and the goal is to estimate its mean. While producing a single estimate is impossible, these algorithms return a list of size $1 / \alpha$, one of which is guaranteed to be accurate. In this paper, the authors essentially extend the SEVER framework to the list-decodable setting. However, this requires some novel technical contributions. A naive implementation would lead to exponential growth in the number of estimates (a list of size $l$ would have $l / \alpha$ many elements in the next iteration if each of its elements were queried and updated with the $1 / \alpha$ resulting gradients). Instead the authors introduce a novel technical tool that they term location estimation which when given samples from $z + \xi$ and $z' + t + \xi$ for some unknown $t$ (and $z$ and $z'$ have bounded covariance) can estimate $t$. With this tool, the algorithm starts by first generating $1/\alpha$ gradients at $0$ and initializing $1 / \alpha$ candidates each corresponding to one of the estimates. Then, each element of the list, $x$, is queried to produce gradient estimates. The location estimate procedure is then run on gradient estimates from $x$ and $0$ to essentially estimate $\nabla f(x) - \nabla f(0)$. Finally, from this each candidate is updated by estimating $\nabla f(x)$ using the particular estimate of $\nabla f(0)$ it corresponds to.

Overall, this is a really nice paper studying an interesting problem. My one concern is in the assumptions made in the paper. For instance, the assumptions don't capture the canonical estimation problem of list-decodable mean estimation. Here, one assumes that the true distribution has covariance bounded in spectral norm whereas this paper essentially assumes a bound on the expected squared length of a data point which could be larger by a factor of $d$. It would be interesting to see if these results could be extended to setting with weaker assumptions. Can we obtain similar results with $\mathbb{E} [(\nabla f(x, \gamma) - \nabla f(x)) (\nabla f(x, \gamma) - \nabla f(x))^\top] \prec \sigma^2 I$ as opposed to the stronger assumption in this paper of $\mathbb{E} [\|\nabla f(x, \gamma) - \nabla f(x)\|^2] \prec \sigma^2$ used here.


**Strengths:**

See main review

**Weaknesses:**

See main review

**Questions:**

See main review

**Limitations:**

Yes

---

> ### Author Rebuttal · Authors · 2023-08-10
>
> We first give a common response, and then addresses the specific concerns of the reviewer.
>
> **Common Response**
>
> To summarize, our paper investigates a generalization of the heavy-tailed stochastic optimization problem.
> As highlighted by reviewers M73n, NDsr, and 2f9g, our work addresses a challenging and well-motivated scenario with minimal noise assumptions.
> It introduces a novel perspective on the connection between stochastic optimization and problems such as list-decodable mean estimation and location estimation,
> employing an innovative efficient reduction (reviewer 2f9g and FMyo).
> A significant contribution of our work is the resolution of the noisy location estimation problem. The proof of our algorithm’s guarantee is novel and technically involved (reviewers 2f9g and NDsr).
> Notably, our method produces a single candidate for a sufficiently small constant fraction of outliers, allowing for a direct comparison with previous approaches in stochastic convex optimization.
>
> We will fix the typos and address minor comments mentioned by the reviewers in the final version of the manuscript.
> We now address more specific concerns of the reviewer.
>
> **Response to Reviewer M73n**
>
> Thank you for your insightful and detailed review.
>
> With regard to your concern about the disconnect between the noise models in robust statistics and the noise in the paper, the assumption we make is common in the optimization literature (see, for e.g. [1,2]). In this setting, these works assume that the squared norm of the deviation satisfies bounded variance.
>
> Indeed, this setting is easier than the usual assumptions that are used in the robust statistics literature. However, even in this setting, we demonstrate a lower bound on the list size that is exponential in the approximation error (see L835 - L855 of the supplementary material).
>
> Extending this to the more general setting is an interesting future direction.
>
> [1] V. V. Mai and M. Johansson. Stability and convergence of stochastic gradient clipping: Beyond lipschitz continuity and smoothness. International Conference on Machine Learning, 2021.
>
> [2] E. Gorbunov, M. Danilova, and A. Gasnikov. Stochastic optimization with heavy-tailed noise via accelerated gradient clipping. Advances in Neural Information Processing Systems, 33: 15042–15053, 2020.

---

> > ### Comment · Reviewer_M73n · 2023-08-18
> >
> > Thank you for your response. I will retain my current evaluation.

---

### Official Review · Reviewer_NDsr · 2023-07-07

**Soundness:** 4 excellent
**Presentation:** 4 excellent
**Contribution:** 3 good
**Rating:** 7
**Confidence:** 1

**Summary:**

This paper studies robust first order optimization in a challenging setting where noise may be unbounded, a setting that arises often for real world optimization problems. Because this problem is intractable in general, one needs to make plausible assumptions on the noise, that are realistic on one hand but allow for efficient analysis.

The noise model proposed here allows for noise to be unbounded, and introduces two simple constraints:

1. The unbounded noise when computing a gradient is *oblivious* in the following sense: there are two noise components, one that is well-behaved (zero mean and bounded variation), and one that is unbounded but oblivious/indpendent of both the location in which gradient is computed and the value of the well-behaved part of the noise.</li>

2. We assume the unbounded noise has probability bounded away from 0 to be equal to zero (i.e., to not exist at all).

It turns out that these two relatively weak conditions allow for efficient robust first order optimization. Specifically, these conditions allow for list-decodable robust optimization, where the goal is to output a list of candidate outputs where at least one should be a good approximation of the correct optimization outcome. The main technical result shows how to solve this problem by reducing it to list-decodable mean estimation, a problem that enjoyed substantial progress in recent years. The authors also show a reduction in the opposite direction. A substantial component in the technical analysis is a procedure that the authors develop for location estimation in an appropriate noisy setting.

**Strengths:**

1. Interesting and important goal, of better understanding the beyond worst case landscape for (first order) optimization.

2. Writing is very clear and relatively easy to follow for me (a non-expert outsider).

3. The assumptions required for the analysis are weak and seemingly realistic.

**Weaknesses:**

1. The technical novelty is perhaps somewhat limited, the work relies heavily on reductions to existing results in robust mean estimation.



**Questions:**

Comment: my review is a low-confidence one (as a non expert in the field) and I may have missed central points in the paper, so may update the score after subsequent reviewers and authors discussions.

---

> ### Author Rebuttal · Authors · 2023-08-10
>
> We first give a common response, and then addresses the specific concerns of the reviewer.
>
> **Common Response**
>
> To summarize, our paper investigates a generalization of the heavy-tailed stochastic optimization problem.
> As highlighted by reviewers M73n, NDsr, and 2f9g, our work addresses a challenging and well-motivated scenario with minimal noise assumptions.
> It introduces a novel perspective on the connection between stochastic optimization and problems such as list-decodable mean estimation and location estimation,
> employing an innovative efficient reduction (reviewer 2f9g and FMyo).
> A significant contribution of our work is the resolution of the noisy location estimation problem. The proof of our algorithm’s guarantee is novel and technically involved (reviewers 2f9g and NDsr).
> Notably, our method produces a single candidate for a sufficiently small constant fraction of outliers, allowing for a direct comparison with previous approaches in stochastic convex optimization.
>
> We will fix the typos and address minor comments mentioned by the reviewers in the final version of the manuscript.
> We now address more specific concerns of the reviewer.
>
> **Response to Reviewer NDsr**
>
> Thank you for taking the time to review our paper.
>
> Indeed as you point out, a large part of our contribution is connecting the robust statistics literature to stochastic optimization, which involves developing novel efficient reductions between existing results in robust statistics and stochastic optimization for our setting.
>
> That said, we would also like to emphasize that our location-estimation algorithm is novel and technically involved (algorithm 4, L608-609 in the supplementary material).
>
> Please look at Section C of the supplementary material for more details.

---

### Official Review · Reviewer_FMyo · 2023-07-26

**Soundness:** 3 good
**Presentation:** 2 fair
**Contribution:** 3 good
**Rating:** 4
**Confidence:** 3

**Summary:**

This paper introduces a new setup for stochastic optimization, where in addition to random observation noise, the stochastic gradient may be subject to independent oblivious noise. This noise might not have bounded moments and isn't necessarily centered. The authors propose a Noisy Location Estimation approach that estimates the gradient difference between two points, specifically \nabla f(x_t)-\nabla f(x_0). As such, they maintain robust estimations of the gradient at all points {x_t} as long as there is a reliable estimation of \nabla f(x_0).

**Strengths:**

The new setup for oblivious noise introduced in the work is plausible, and the authors effectively discuss its relation to existing research. The Noisy Location Estimation proposed by the authors provides an innovative way to estimate gradient differences accurately, reducing the stochastic optimization problem to a mean estimation problem, which seems simpler in the setting. Also, as shown by the authors, the reverse of the reduction holds by simple arguments.

**Weaknesses:**

The paper's presentation, particularly in the technical sections, lacks clarity.

1. Definitions should be more precise and self-contained. For instance, the work seems to require that oblivious noise be independent of the noisy gradient, but Definition 1.1 doesn't explicitly state this. In Definition 1.3, phrases like "sufficiently small constant" are too vague.
2. The methodology for mean estimation (Fact 2.1), isn't discussed in the main body. A brief discussion may be helpful.
3. The "Rejection Sampling" discussion on page 5 is difficult to follow and potentially misleading. From my understanding, the core intuition is to identify a large enough domain of size $i$, such that $i$ times the conditional expectation is robust and stable upon shifting the domain.

**Questions:**

1. In Line 227, it appears that [i- 4 · 12, i + 4 · 12) almost fully contains [i - 4 · 12, -i + 4 · 12)]. If that's the case, why is there a need for a \cup operation?
2. In Section 5, why is the exponential dependence on 1/\eta for the size of the list unavoidable?
3. Is it a requirement for the Noisy Location Estimation that alpha > 0? I didn't delve into the proof details, but it seems that if you're considering the conditional expectation, it might not require alpha > 0. Can you clarify this?

**Limitations:**

As discussed in Weakness.

---

> ### Author Rebuttal · Authors · 2023-08-10
>
> We first give a common response, and then addresses the specific concerns of the reviewer.
>
> **Common Response**
>
> To summarize, our paper investigates a generalization of the heavy-tailed stochastic optimization problem.
> As highlighted by reviewers M73n, NDsr, and 2f9g, our work addresses a challenging and well-motivated scenario with minimal noise assumptions.
> It introduces a novel perspective on the connection between stochastic optimization and problems such as list-decodable mean estimation and location estimation,
> employing an innovative efficient reduction (reviewer 2f9g and FMyo).
> A significant contribution of our work is the resolution of the noisy location estimation problem. The proof of our algorithm’s guarantee is novel and technically involved (reviewers 2f9g and NDsr).
> Notably, our method produces a single candidate for a sufficiently small constant fraction of outliers, allowing for a direct comparison with previous approaches in stochastic convex optimization.
>
> We will fix the typos and address minor comments mentioned by the reviewers in the final version of the manuscript.
> We now address more specific concerns of the reviewer.
>
> **Response to Reviewer FMyo**
>
> Thank you for taking the time to review our paper and for appreciating the innovative reduction.
>
> We address your concerns below, most of which have been addressed in the supplementary material:
>
> 1. **“Sufficiently small constant”**:
> This is common terminology in the learning theory literature. It means “there is a sufficiently small universal constant”. In our case, a constant of $\min{½, \eta^2}$ suffices, where $\eta$ is the final intended approximation error. Prior work in algorithmic robust statistics is able to perform efficient robust mean estimation tolerating up to half the samples being corrupted while achieving a guarantee of $\sigma \sqrt{\epsilon/(1-2\epsilon)}$, where $\epsilon$ is the fraction of corrupted points (see, for e.g. exercise 2.10 in [2], or [1]).
>
>
> 2. **Definition of oblivious noise**:
> Indeed, the noise is independent of $x$ as well as $\gamma$ as mentioned on lines L5-L8 of the abstract.
>
>
> 3. **Methodology for list-decoding**:
> We use a list-decodable robust mean estimation subroutine in a black-box manner in the main body of our paper. (That is, one can use any efficient algorithm with this performance guarantee in our setting; prior work has developed such algorithms, see, e.g., [3, 4]). That said, please see the algorithm described on L821-L826 in the supplementary material.
>
>
> 4. **Rejection sampling procedure hard to follow**:
> The reviewer’s understanding of the rejection sampling procedure is accurate. We will clarify any potentially confusing phrasing.
>
>
> 5. **Line 227**:
> Yes, this is a typo. The second interval is $[-i - 4 \cdot 12\sigma, -i+4 \cdot 12\sigma)$. The second term is “$-i$”.
>
>
> 6. **Exponential dependence on $1/\eta$**:
> On L835-L855 of the supplementary material, we provide an information-theoretic lower bound demonstrating that the exponential dependence is inherent. Specifically, we show the existence of $\exp(1/\eta)$ different pairs of inlier distributions and oblivious noise distributions, each inlier distribution having a different mean, such that the sum of the inlier distribution and the oblivious noise distribution is exactly the same in every case.
>
>
> 7. **$\alpha > 0$ requirement**:
> There must be some concentration that the oblivious noise satisfies, since for instance it is not possible to efficiently distinguish between $[-R, R]$ and $[-R+1, R+1]$ as $R \rightarrow \infty$. In our algorithm this appears in our noisy location estimation, which runs in two phases. The first step of this is a rough estimate, whose error goes as $O(\sigma \alpha^{-½})$ – and this is tight, as shown in the proof of the rough estimation –  the second step is to improve this to a much finer estimate (see L614-L617 of the supplementary material). For the rough estimation to work, we will require $\alpha > 0$.
>
> [1] B. Zhu, J. Jiao, and J. Steinhardt. Robust estimation via generalized quasi-gradients. Information and Inference: A Journal of the IMA 11.2  2022.
>
> [2] I. Diakonikolas, and, D. Kane. Algorithmic High-dimensional Robust Statistics.
>
> [3] M. Charikar, J. Steinhardt, and G. Valiant. Learning from untrusted data. Proceedings of the 49th Annual ACM SIGACT Symposium on Theory of Computing. 2017.
>
> [4] I. Diakonikolas, D. Kane and D. Kongsgaard. List-Decodable Mean Estimation via Iterative Multi-Filtering. NeurIPS 2020

---

> > ### Comment · Reviewer_FMyo · 2023-08-21
> >
> > Thanks for your response. I raise my score by one point. However, I still believe the manuscript's writing quality falls short of what I would expect for a NeurIPS submission.

---

### Decision · Program_Chairs · 2023-09-21

**Decision:**

Accept (poster)

**Comment:**

The paper has received mostly favorable reviews, with reviewers appreciating the novelty of the setting and the proposed method. However, the reviewers have also noted several issues with the writing, and most notably a somewhat careless use of random variables to denote distributions, without clarifying whether repeated instances of the same greek letter indicated a fresh sample from the same distribution or not. (In addition to the instances of this issue pointed out by reviewers, I also noted that $\zeta_i$ in line 2 of Alg 3 is not the same as $zeta_i$ in line 3 of that algorithm.)

After additional consideration, in my opinion the paper passes the bar for acceptance to NeurIPS. For the camera-ready submission, please take care to clarify the issue outlined above as well as all the other points that came up during the review process.